# Factors influencing the timely completion of construction projects in Sri Lanka

**Nadeesha Abeysinghe**[1], **Ruwan Jayathilaka**[2]*

1 MBA Candidate and a Civil Engineer, SLIIT Business School, Sri Lanka Institute of Information Technology, Malabe, Sri Lanka, 2 Head—Department of Information Management, SLIIT Business School, Sri Lanka Institute of Information Technology, Malabe, Sri Lanka

* ruwan.j@sliit.lk

**Data Availability Statement:** All relevant data are within the manuscript and its Supporting Information files (S2 Appendix. Data File).

**Funding:** The authors received no specific funding for this work.

## Abstract

Timely completion is a crucial factor for the success of a construction project, especially in the Sri Lankan context. This study aims to identify the most influential factors that affect the timely completion of construction projects in Sri Lanka. Thirty-nine factors were identified through a comprehensive literature review and experts' opinions. A questionnaire incorporating the 39 project delay factors was distributed among 163 Civil Engineers, and responses were obtained. Random sampling method was adopted to select the sample. The Relative Importance Index (RII) analysed and ranked the project delay factors. The top ranked significant project delay factors were identified as shortage of skilled subcontractors/suppliers, shortage of labourers (Skilled, semi-skilled, unskilled), financial difficulties of contractors, delay in delivering materials to the site, and Covid-19 pandemic situation. According to the main three respondent types, i.e., clients/owners, contractors and consultants, the contractor related factors was the key group among others that delay a construction project. The scientific value of the study includes assisting the Sri Lankan construction industry to identify the factors affecting the timely completion of construction projects, and developing mitigation methods and strategies. Also, the stakeholders could duly schedule the construction work by identifying areas that need more attention. The contribution of this study would assist stakeholders to adopt a proactive approach by identifying mistakes on their part and minimising potential issues that lead to construction project delays in Sri Lanka.

## Introduction

The construction industry is a key contributor to the Sri Lankan economy. Currently, the construction industry accounts for 7.1% of the gross domestic product (GDP) in Sri Lanka. Also, over 600,000 labourers are currently employed countrywide construction sites in Sri Lanka. Annually, the construction sector generates approximately LKR 4.2 billion of revenue [1].

Due to the covid-19 pandemic, every sector had its share of downfall. Nevertheless, with the slow-paced economic recovery, the construction industry too is picking up slowly but steadily. At the time of writing, many construction projects have recommenced work and are rising slowly. Many massive infrastructure projects are currently under development, such as the

**Competing interests:** The authors have declared that no competing interests exist.

Port City Development Project, the extension of the Southern Expressway from *Matara* to *Hambanthota*, the Central Expressway Project etc. Apart from these, the country has many ongoing major housing, residential and commercial projects. Under these circumstances, the Sri Lankan construction industry is expected to undergo massive growth within the next 15 to 20 years [2].

The nature of the construction industry can be considered as uncertain. Construction projects differ from each other depending on the project size, project objectives, project duration, etc. Every project is unique on its own and no project has the same characteristics. Even though the construction projects nowadays use advanced and new project management theories and technologies, the delay in the completion of projects cannot be mitigated [3].

The time deviation of a construction project can be defined as the difference between specified project duration and the real project duration. There can be three types of time deviations in a construction project. Firstly, is a negative deviation, where the real duration is less than the specified duration. Secondly, there is the no particular deviation type, where the specified duration and the real duration are the same. Thirdly, is the positive deviation, where the real duration is greater than the specified duration. This positive deviation is also known as the time overrun, where the delays in the project completions occur. When the delay period is long, consequently the effects will also be greater/significant, which can exert a negative impact on the project. For the successful completion of a project, cost, quality as well as time, should be properly utilised [4].

Therefore, timely completion is one of the crucial factors for the success of any construction project. At the initial stage, a project should be well planned to be delivered within the specified time range. In Sri Lanka, it is common for most construction projects to get delayed for various reasons [5]. Construction project activities revolve around the client, contractor, and consultant, who are its main stakeholders. Therefore, the key stakeholders should properly plan, schedule, and monitor each phase and every key activity of the project throughout the process. This is because when a construction project is delayed, the parties mentioned above will also get severely affected [6]. The client will lose their revenue as the project could not be available for business. When the project gets delayed, it means that contractors, too require a longer time duration than initially estimated. Accordingly, the contractors will face financial difficulties, additional charges, penalties for time overrun etc., to pay wages for the labourers, materials, and equipment for an extended time. As such, costs will push up for the remaining activities while revenue will remain almost unchanged in line with estimates. Further, delayed projects constrain securing new construction projects as well as losing the earning potential. In other words, a delayed project means more expenses, narrow profit margins, loss of credibility and reputation, and loss of future revenue [7].

Various factors influence a construction project. Resource-related issues are also causing major problems for any construction project. Resources include human resource, materials, equipment, etc [8]. The procurement process is a vital part of the project that should be duly completed within the specified project duration. The external environment can adversely affect project completion. The external stakeholders, such as the government, regulatory bodies, the public, etc., can arouse issues for project completion in numerous ways. These include changes in regulations, and inefficient handling of approvals by govt. authorities [9]. Therefore, that time overruns can occur due to various factors. This study will attempt to identify the most crucial factors that can influence a construction project duration in Sri Lanka.

The objective of this study is to identify the most significant factors that can affect the timely completion of a construction project in Sri Lanka. The scientific value of the study can be elaborated by identifying the difference between the present study and similar studies which have been conducted previously on the Sri Lankan construction industry. This can be explained in

four ways. Firstly, the study has analysed the responses given by the main three stakeholders of a construction project, namely, client/owner, contractor, and consultant. Based on their opinions the most influential project delay factors were identified with respect to each type of stakeholder. It is a must to understand perspectives of each stakeholder on construction project delays, and in reality, how each party play the blame game.

Secondly, the study was done based on factors related to client/owner, contractor, consultant, resources, and external factors. Based on the results, the most important factor in each group was identified. This will be useful for the stakeholders to identify which factors they should pay more attention to, and factors to be considered when setting priorities under each category. Thirdly, the most influential group of factors, which could affect the timely completion of a construction project was identified. Since the construction sector plays a major role in contributing to the country's economy, the findings of the study help policymakers gain valuable insights into construction project delays.

Finally, the study will enable to set up a platform for Civil Engineers in Sri Lanka for knowledge sharing and collaboration with experts and construction players. This type of approach will assist them in sharing their opinion and addressing existing and potential issues etc., regarding construction project delays in Sri Lanka.

## Literature review

Many studies have been conducted in various parts of the world to identify the factors causing time and cost overruns in construction projects. Some studies have ranked the factors according to their effectiveness on the project duration, while others have suggested mitigation methods to overcome delays. Several publications were critically analysed to understand better and identify different perspectives on how past researchers have addressed the selected research area for this study. These findings will be useful in decision making to choose the most suitable research path and identify the research gap. Fig 1 shows how the literature search was carried out in a step-by-step approach.

The literature review for this study consists of 40 published articles identified using a comprehensive literature search. Authors referred to reputed search databases such as Emerald insight, Science direct, Taylor & Francis online, Wiley online and Springer for this purpose. Most studies focus on analysing the major project delay factors based on the different stakeholders' perspectives. However, the critical project delay factors did not have a noticeable relationship among other continents. Various financial situations, availability of resources, government regulations etc., have led to the difference in delay factors between continents [10]. Therefore, to provide a better understanding, these articles were recategorised into three based on the continent where the authors have addressed the research problem. Therefore, the articles were categorised as factors affecting the project delivery in countries in Asian, African and European regions.

**Factors affecting construction project completion in Asian countries.** In Malaysia, Alaghbari, Razali A. Kadir [11] found that the most effective factors related to project delay have been linked with the contractor. The study was conducted to identify the major causes of delay in building system construction projects in Malaysia, where the data were collected from several construction parties. Here, the most effective factors related to project delay have been occurred by the contractor, and among these, financial problems faced by the contractor play a major role in delaying the construction projects. Zailani, Ariffin [12] found the relationship between possible causes of delay and construction project performance. Out of the 1,322 registered construction project companies in Malaysia, data were collected from 204 companies. The results showed that factors related to coordination, resources and environment were the

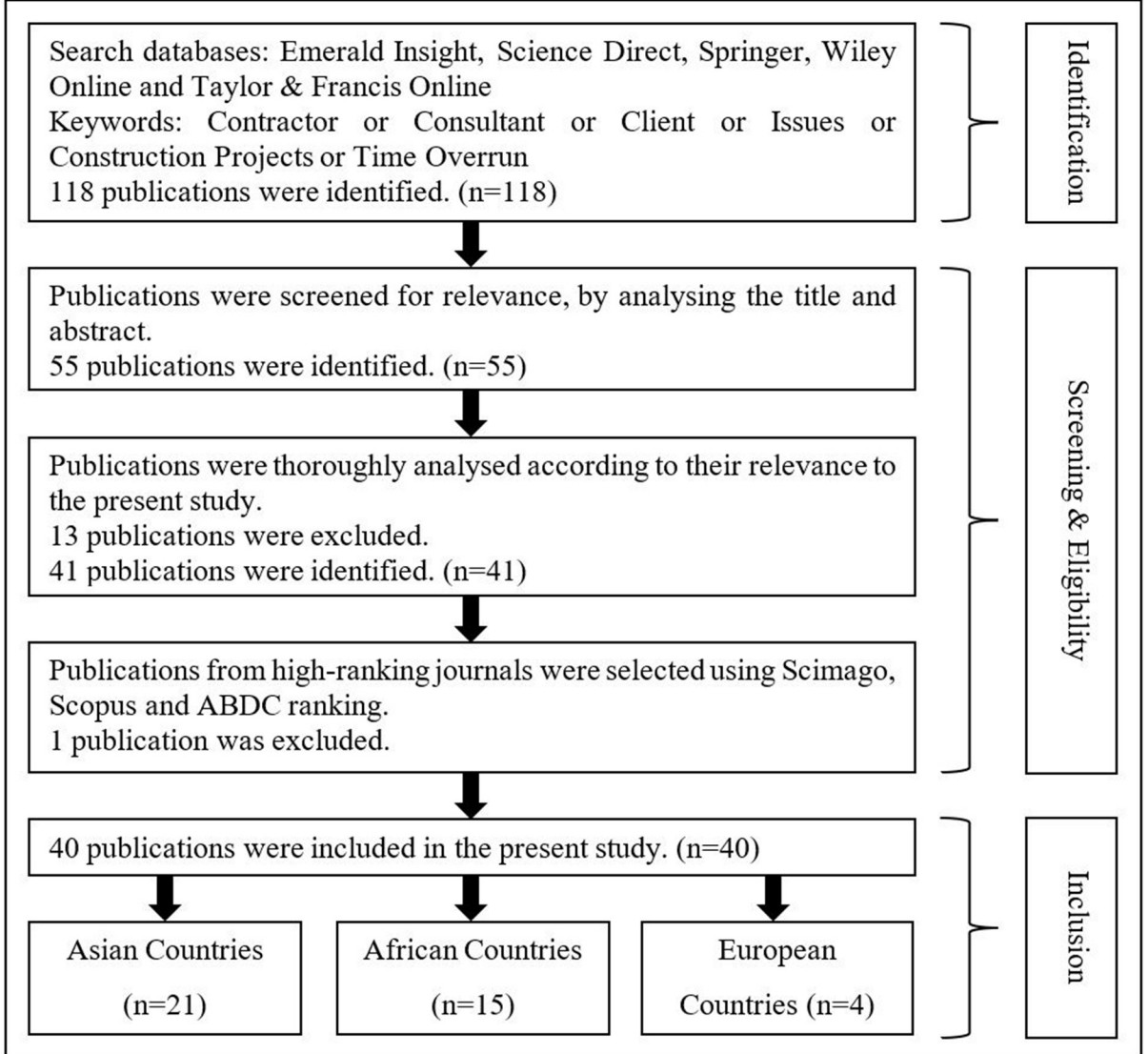

**Fig 1. Literature search flow diagram.** Source: Based on authors' observations.

most effective in minimising delays. The study also suggested that delays could be mitigated by supplier development as well as the project flexibility and visibility. In another research conducted in Malaysia, Yap, Goay [13] found that the constant design changes by the client during construction, lack of experienced supervisors and subcontractors, financial difficulties of contractors and poor scheduling and planning led to the construction project delays. Enshassi, Al-Najjar [14] found that the material-related factors and the labour strikes affected the construction project delays in the Gaza strip. This study was conducted in 2009 to identify the key variables causing construction project delays from the perspective of contractors, consultants, and clients. Mahamid [15] developed a risk matrix which depicts the possible time overrun factors in Palestine. This study was limited to road construction projects in the West Bank of Palestine, in which, the factors reflected only from the client's point of view. The risk matrix developed in this study could be used to identify the impact of each delay factor and its probability of occurrence. Gardezi, Manarvi [6] conducted a study in Pakistan in which the construction

project delay factors were ranked using the Relative Importance Index (RII). It was found that the external factors related to a construction project could highly affect the timely delivery of the project.

In 2016, a study on construction project delay factors in Gulf Countries Construction industry, Elawi, Algahtany [16] intended to find the construction parties responsible for each delaying factor. The results suggested that the most effective project delaying factor was land acquisition and often, construction project delays occurred on the client's part. Alsuliman [17] researched factor analysis, where the results developed an equation to calculate the actual project duration. This was useful and practical for the Saudi Arabian government to take necessary measures to mitigate public construction project delays. Assaf, Hassanain [18], proved that changes in orders given by clients during the construction, contractor delays and design errors were the most severe project delay factors in the Eastern Province of Saudi Arabia.

In a study conducted in Jordan, Al-Hazim, Salem [19] intended to identify the time overrun factors in public infrastructure projects in Jordan. The study conducted using final reports obtained from 40 infrastructure projects completed between 2000 and 2008 revealed that unfavourable site conditions and weather conditions were the most severe project delay factors. In 2019, another research investigated the delay factors in public construction projects in Jordan. It is noteworthy that the clients and consultants in Jordan were concerned about the delays that occurred by themselves, while the contractors were more concerned about the delays that occurred by the clients Ahmad, Ayoush [20].

In Cambodia, Durdyev, Omarov [8] identified the time overrun factors in residential building construction projects in the country. The factors were ranked according to their importance where material shortage, unrealistic project durations and the lack of skilled labour were the most severe factors to delay the residential building projects. Mpofu, Ochieng [7] claimed that unrealistic project durations and decreased labour productivity have affected the timely completion of construction projects in the United Arab Emirates. It was suggested that all construction parties, including client, consultant and contractor should reorganise their working patterns to successfully complete a construction project. In Iran, a model was developed by Parchami Jalal and Shoar [21] to identify the most effective delaying factor for construction project completion. Here, the factors related to client were the most effective while the external factors were the least effective. Shahsavand, Marefat [22] revealed that the main delay factors in construction projects in Iran were caused by contractors, labourers, clients, and equipment. The relationship between the project delay factors in Iran was analysed by Jahangoshai Rezaee, Yousefi [23]. It was identified that the technical faults during construction and unrealistic workload estimation were the main causes of construction project delays in Iran.

Wang, Ford [24] identified the primary causes of project delays in construction projects in China as payment delays, poor performance by subcontractors and communication problems. In a similar study, Prasad, Vasugi [25] claimed that delay in settlement claims and the financial difficulties of contractors and clients were the most significant delay factors in India. No difference was reported between the delay factors caused in design-build projects and design bid-build projects. Hoque, Safayet [10] confirmed that payment delays and errors during construction were the most significant delay factors in Bangladesh. Although ranked based on their importance, the effect of delay factors on the overall project duration was not analysed.

In Sri Lanka, Santoso and Gallage [5] identified that the contractor-related factors were the most influential factors in delaying large construction projects. The sample size taken for this study was relatively small compared to many construction companies in Sri Lanka.

**Factors affecting the construction project completion in African countries.** In 2014, a study was conducted in Egypt to examine the construction project delay factors, where poor planning and scheduling and unfavourable soil conditions were the most critical delay factors

[26]. In another study by Aziz and Abdel-Hakam [9], contractor-related factors were the most influential factors in Egypt. Some scholars developed models and frameworks on factors concerning construction project delays. In 2020, El-Rasas and Marzouk [27] analysed the causes of residential construction project delays and developed a fuzzy model to determine the probability of delay. Elhusseiny, Nosair [28] developed a systemic processing framework in which the most influential delay factors for Egyptian construction projects were slow decision making, changes in the scope of work and payment delays.

Amoatey, Rolf [29] found that the factors related to financing affect the completion of the public housing construction industry in Ghana. The following year, 86.6% of construction projects in Ghana experienced delays in completion. The samples were obtained only from public school projects. In 2017, the projects related to the education sector in Ghana were analysed. In this study, Famiyeh, Amoatey [30] identified financial problems as well as unrealistic project durations as the most effective delay factors in Ghana. Asiedu & Gyadu-Asiedu, 2019 developed a baseline model to analyse the time overrun of construction projects in Ghana, which was more effective in predicting time overrun than using a multiple regression model. This study also focused on public-school projects using data obtained from the school construction projects completed between 2010 and 2013.

In a South African-based study, Mukuka, Aigbavboa [31], the construction project delay adversely affects the construction parties on a personal level and also the construction company's reputation. In 2017, corruption, lack of resources, increased material prices and poor site management were the typical causes of delay in construction projects in Ethiopia [32]. Abebe, Germew [33] used Pareto analysis to analyse the delay factors in which lack of utilities and finance-related factors were the main delay causes. The drawback of this study was that the sample size was smaller when compared with the previous study conducted in Ethiopia.

In Nigeria, the quality control of the construction project, financial problems, unfavourable site conditions and fluctuation of material prices were the main causes of delay [34]. Fashina, Omar [35] found that the contractor-related factors are the most effective in delaying the road and building projects in Hargeisa. The study on both public and private construction projects enhanced the significance of the study. Although Mwamvani, Amoah [36]'s quantitative study suggested methods for construction project delays, it was limited to a single organisation in Malawi, hence the findings could not be generalised.

**Factors affecting the construction project completion in European countries.** Agyekum-Mensah and Knight [37] analysed the construction project delays in the United Kingdom using a qualitative method. The study revealed that poor planning and management, and poor communication and resource management were the most influential delay factors. In 2018, Zidane and Andersen [38] analysed the major Norwegian construction projects, where constant design changes, payment delays, poor site management, and financial problems were the main causes of delay. This study included the participation of 202 respondents from the construction industry. Arantes and Ferreira [39] used factor analysis to find the causes of delays in construction projects in Portugal. Poor planning, consultant performance and poor site management were the most influential delay factors. The sample size was relatively small (94) compared to the population size 2,100. In Denmark, Lindhard, Neve [40] claimed that the construction design and connecting workers and labour force were the most effective delay factors. The study only focused on resource-related factors.

According to the literature, the focus of many past studies was to analyse the factors affecting the timely completion of construction projects. These studies were sometimes conducted only within a specific type of construction project, such as road, building, infrastructure, etc. Also, some studies analysed either public construction projects or private construction projects and in some countries, the study scope was limited to certain areas. Therefore, for

comprehensiveness and to fill the research gap, the present study has included every type of construction project, not limiting to projects in some regions of the country. Most of the past studies used the RII to rank the delay factors, as it was accepted as an accurate method to calculate the importance of each factor. Therefore, RII has been used to rank the project delay factors in the present study. Furthermore, the delay factors were analysed based on the type of respondents and the group of factors. The major group of factors that could affect the timely completion of a construction project was also identified.

The subsequent sections of this paper outline the methodology followed by study results, conclusion, and finally reaching the recommendation and policy implications.

## Methodology

This section presents the methods and techniques used for data collection and analysis. Fig 2 depicts the flowchart of the research process used for the study.

### Questionnaire design

As the research strategy, survey strategy was adopted for the study where a questionnaire was developed in which the project delay factors were included. The respondents were able to rank each factor according to their significance, with the use of a questionnaire. The questionnaire was developed using 39 project delay factors, which can affect the timely completion of a construction project in Sri Lanka. These factors were identified through a critical literature review and from experts' opinion. Ten experts were consulted for this purpose. Four authors who have conducted the same type of research studies in the past few years, were contacted to identify the global impact [5, 13, 35, 39]. Six Civil Engineers from the Sri Lankan construction Industry were contacted to identify the national impact.

The demographic data of the respondents were collected under general information at the beginning of the questionnaire, while the rest of the content was divided into five sections. These five sections consisted of eight client/owner related delay factors, followed by eight

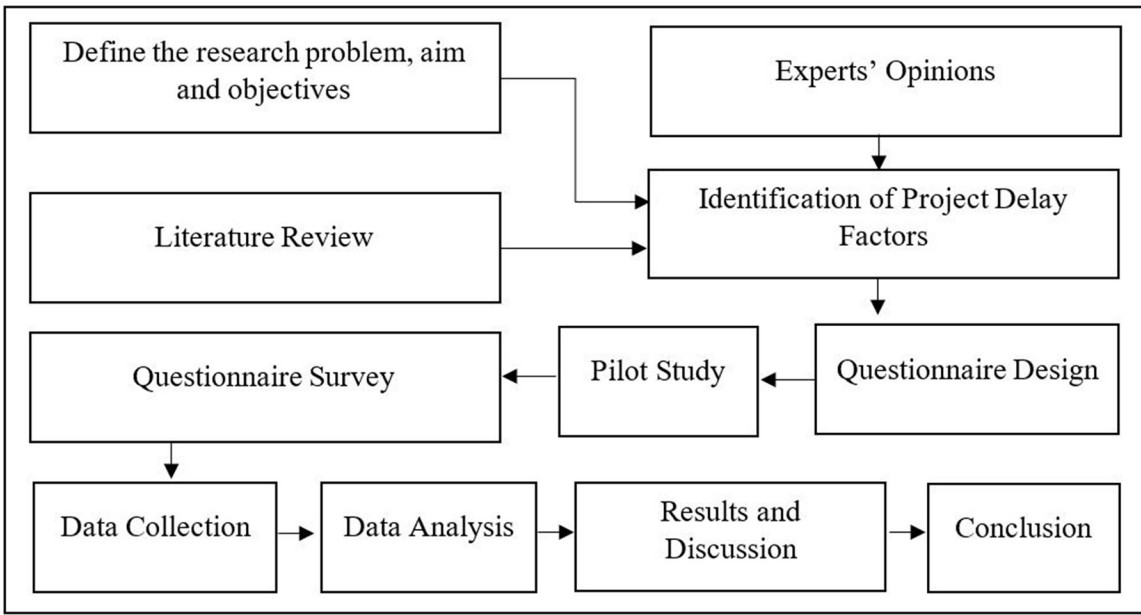

**Fig 2. Flowchart of the research process.** Source: Based on authors' observations.

contractor related delay factors, eight consultant related delay factors, eight resource related delay factors and finally seven general external delay factors, which include soil conditions, weather conditions, government regulations etc. A five-point likert scale was used to collect data for each delay factor which ranged from 1 (Very low significance) to 5 (Very high significance). The questionnaire with the finalised project delay factors is in the S1 Appendix.

## Data collection

The questionnaire was distributed among Civil Engineers via electronic mail and manually. Qualified Civil Engineers were selected as the population, since they are engaged in the construction work under all three stakeholder groups namely, client/owner, consultant and contractor. The list of qualified Civil Engineers was obtained through the Construction Industry Development Authority (CIDA) website. A pilot study was carried out by distributing the questionnaire among 10 Civil Engineers. Then, the final questionnaire was distributed to 1,416 respondents selected using a random sampling method from 2,716 Civil Engineers included in the list. One hundred sixty-three responses were collected, of which, 28.8% were from clients/ owners, 28.2% were from contractors, 39.9% were from consultants and 3.1% were from other respondents (Civil Engineers who are currently retired or engaged in other work). The data file is presented in the S2 Appendix.

## Data analysis

According to Kometa, Olomolaiye [41], RII method was used to determine the importance of each delay factor. The data collected through the questionnaire were used to calculate the RII, mean value and standard deviation for each project delay factor. The following Eq 1 was used to calculate the RII.

$$RII = \frac{\sum_{i=1}^{5} wx}{A \times N} \tag{1}$$

where,

 *RII* = Relative Importance Index
 *A* = 5 (Highest weight)
 *N* = Total count of respondents
 *w* = 1 to 5 (weights given to each factor by each respondent)
 *x* = frequency of responses given to each factor
 Weights were assigned to each factor by each respondent (1 = Very low significance, 2 = Low significance, 3 = Average significance, 4 = High significance, 5 = Very high significance) were multiplied by the frequency of responses given to each factor, and the total sum of those two values was calculated. The result was divided by the multiplication of the highest weight (5) and the total count of respondents. The ranking of the delay factors was done using the RII. Overall rankings were calculated by combining the responses given by all the respondents.

 The responses collected through the open-ended questions included in the questionnaire were analysed using an online word cloud generator for visualisation. The most commonly used word by the respondents is the one which appears to be the largest within the cloud.

## Variable analysis and validity of the questionnaire

The internal consistency of the factors was analysed using Cronbach's alpha coefficient [42]. The pilot study, which consisted of 10 responses, produced a coefficient of 0.9157. The coefficient for the present study, which consisted of 163 responses was found to be 0.9505.

According to Gliem and Gliem [43], if Cronbach's alpha has a value equal to or greater than 0.9, then the internal consistency of the factors can be considered excellent. Therefore, the internal consistency of the collected data was excellent.

## Results and discussion

### Demographic characteristics

The demographic characteristics of 163 respondents are shown in Table 1.

According to Table 1, most respondents (82.8%) were male. Among these, 58.9% of the respondents were above 40 years, 55.2% of the respondents had a Master's degree, while 42.3% had a Bachelor's degree as their highest educational qualification. The majority of the respondents (46.6%) had work experience of more than 20 years in the Civil Engineering field. Based on the type of organisation, 28.8% of the respondents were employed at infrastructure

**Table 1. Demographic characteristics of the respondents.**

| Demographic Characteristics | Percentage (%) |
|---|---|
| **Gender based** | |
| Male | 82.8 |
| Female | 17.2 |
| **Age based** | |
| 20–29 years | 28.2 |
| 30–39 years | 12.9 |
| 40 years and above | 58.9 |
| **Highest educational qualification based** | |
| Certificate level | 0 |
| Diploma | 0.6 |
| Bachelor's degree | 42.3 |
| Master's degree | 55.2 |
| PhD | 1.2 |
| Other | 0.6 |
| **Work experience based** | |
| Below 5 years | 27 |
| 5 to 9 years | 1.8 |
| 10 to 19 years | 24.5 |
| 20 years and above | 46.6 |
| **Type of organisation** | |
| Road construction | 26.4 |
| Residential building construction | 11 |
| Commercial building construction | 17.2 |
| Infrastructure construction | 28.8 |
| Water supply/Drainage/Irrigation | 6.7 |
| Other | 9.8 |
| **Type of respondent** | |
| Client/Owner | 28.8 |
| Contractor | 28.2 |
| Consultant | 39.9 |
| Other | 3.1 |

Source: Authors' calculations.

construction projects, while 26.4% were employed at road construction projects. In terms of percentages, clients, contractors, and consultants responded to the questionnaire at 28.8%, 28.2% and 39.9%, respectively.

## Overall ranking

The 39 factors were ranked according to their overall RII. The mean value, standard deviation value, calculated RII and the overall rank for each delay factor are shown in Table 2. The following values are calculated and presented in Sheet 1 in S2 Appendix.

The most significant delay factor was the shortage of skilled subcontractors/suppliers (RII = 0.8294), which fell under contractor-related factors. This result alligned with the results obtained by Yap, Goay (11) and Wang, Ford (22). Shortage of labourers (Skilled, semi-skilled, unskilled) was the second most significant factor (RII = 0.8245) categorised under resource-related factors. This result can be further confirmed by the results obtained by Durdyev, Omarov (6) who conducted a study in Cambodia to identify the time overrun factors in residential building construction projects in the country. These factors seem realistic in the construction industry as most subcontracts/suppliers and labourers are not skilled. The third highest ranked factor was the financial difficulties of contractors (RII = 0.8233), another contractor-related factor. In Malaysia, two research studies (9), (11) confirmed that the financial difficulties of the contractor play a major role in delaying construction projects, while Prasad, Vasugi (23) have also received the same result in the research they conducted in India. Also, delay in delivering materials to the site was the fourth highest ranked delay factor (RII = 0.8098), which was a resource-related factor. Fig 3 shows the graphical depiction of the relationship between the 39 factors and their overall RII.

## Ranking based on the type of respondent

According to the type of respondents, the factors considered in the study do not affect the timely completion of a construction project in the same manner. This means responses are rather subjective. Therefore, it is crucial to identify how different respondents have expressed their unique opinions regarding the delay factors. Table 3 shows the mean value, standard deviation value and the calculated RII for each factor, based on the three main types of respondents, namely, client/owner, contractor, and consultant. The calculation of the following values is presented in Sheets 2,3 and 4 in the S2 Appendix.

Table 4 shows the top ten ranked delay factors for each group of respondents.

According to both clients/owners and consultants, the most significant factor was the financial difficulties of contractors. As noted previously, the shortage of skilled subcontractors/suppliers was the most significant factor according to the contractors, while clients/owners and consultants identified it as the second most significant factor.

According to the contractors, the second high ranked delay factor was the shortage of labourers (Skilled, semi-skilled, unskilled). The clients/owners and consultants ranked this factor the third most significant one. From the contractors' perspective, the third highest ranked delay factor was the delay in delivering materials to the site. Notably, the clients/owners and consultants had nearly the same perspective on the delay factors.

## Ranking of factors in each group of factors

The thirty-nine factors were analysed to find out which factors would be the most significant delay factors in each group. Therefore, the factors were ranked based on their overall RII, within each group of factors.

**Table 2. Mean value, standard deviation, RII and overall rank of each factor.**

| | Types of delay factors | Mean | SD | RII | Rank |
|---|---|---|---|---|---|
| **Client/Owner-related factors** | | | | | |
| Q1 | Changes in design by the client during construction. | 3.4785 | 1.1240 | 0.6957 | 18 |
| Q2 | Slowness of client's decision-making. | 3.6687 | 1.0946 | 0.7337 | 13 |
| Q3 | Unreasonable project duration given by the client. | 3.4663 | 1.1509 | 0.6933 | 21 |
| Q4 | Delay in settling contractor claims by the client. | 3.5951 | 1.0402 | 0.7190 | 14 |
| Q5 | Financial difficulties of the client. | 3.7239 | 1.2033 | 0.7448 | 12 |
| Q6 | Delay in design approvals. | 3.5951 | 1.1687 | 0.7190 | 14 |
| Q7 | Poor communication with contracting parties. | 3.3067 | 1.1074 | 0.6613 | 29 |
| Q8 | Errors in design and specifications. | 3.2331 | 1.2648 | 0.6466 | 32 |
| **Contractor related factors** | | | | | |
| Q9 | Poor planning and scheduling. | 3.9448 | 1.0438 | 0.7890 | 6 |
| Q10 | Shortage of skilled subcontractors/suppliers. | 4.1472 | 0.8835 | 0.8294 | 1 |
| Q11 | Financial difficulties of contractors. | 4.1166 | 0.8635 | 0.8233 | 3 |
| Q12 | Disagreements between the contractor and other parties. | 3.3620 | 0.9351 | 0.6724 | 27 |
| Q13 | Poor site management, monitoring, and control. | 3.8712 | 1.0550 | 0.7742 | 9 |
| Q14 | Errors during construction. | 3.2270 | 1.1016 | 0.6454 | 34 |
| Q15 | Underestimating the project duration. | 3.4724 | 1.0733 | 0.6945 | 20 |
| Q16 | Regular changes of subcontractor's staff. | 3.4785 | 1.0443 | 0.6957 | 18 |
| **Consultant related factors** | | | | | |
| Q17 | Delay in inspections and completed work approvals. | 3.2331 | 1.0633 | 0.6466 | 32 |
| Q18 | Delay in material and payment approval. | 3.4969 | 1.1242 | 0.6994 | 17 |
| Q19 | Errors in contract documents. | 3.0614 | 1.0523 | 0.6123 | 36 |
| Q20 | Constant design changes by the consultant | 3.3681 | 1.2860 | 0.6736 | 26 |
| Q21 | Delay in preparing and approving drawings and design documents. | 3.4479 | 1.1502 | 0.6896 | 22 |
| Q22 | Lack of experienced consultants. | 3.3129 | 1.2096 | 0.6626 | 28 |
| Q23 | Errors in design documents. | 3.1963 | 1.1159 | 0.6393 | 35 |
| Q24 | Poor coordination and communication. | 3.3865 | 1.1776 | 0.6773 | 25 |
| **Resource related factors** | | | | | |
| Q25 | Shortage of labourers. (Skilled, semi-skilled, unskilled) | 4.1227 | 0.9924 | 0.8245 | 2 |
| Q26 | Delay of delivering materials to the site. | 4.0491 | 0.9545 | 0.8098 | 4 |
| Q27 | Poor material handling at the site. | 3.4294 | 1.0539 | 0.6859 | 23 |
| Q28 | Low productivity of labourers. | 3.7423 | 1.0036 | 0.7485 | 11 |
| Q29 | Fluctuation of material prices in the market. | 3.9448 | 1.0497 | 0.7890 | 6 |
| Q30 | Inadequate numbers of equipment. | 3.9141 | 0.9054 | 0.7828 | 8 |
| Q31 | Failure of equipment. | 3.5092 | 1.0679 | 0.7018 | 16 |
| Q32 | Personal disagreements between labourers. | 2.7055 | 1.0360 | 0.5411 | 37 |
| **External factors** | | | | | |
| Q33 | Delay in obtaining permissions/approvals from government. | 3.7791 | 1.0484 | 0.7558 | 10 |
| Q34 | Unknown subsurface conditions. (Soil condition, water table etc.) | 3.4294 | 0.9874 | 0.6859 | 23 |
| Q35 | Bad weather conditions. | 3.2638 | 0.9927 | 0.6528 | 30 |
| Q36 | Accidents during construction. | 2.5215 | 1.0849 | 0.5043 | 39 |
| Q37 | Changes in laws and regulations from the government. | 2.6564 | 1.1297 | 0.5313 | 38 |
| Q38 | Delay in utility services. (Electricity, water etc.) | 3.2454 | 1.1552 | 0.6491 | 31 |
| Q39 | Covid-19 pandemic situation | 3.9509 | 0.9349 | 0.7902 | 5 |

Source: Authors' calculations.

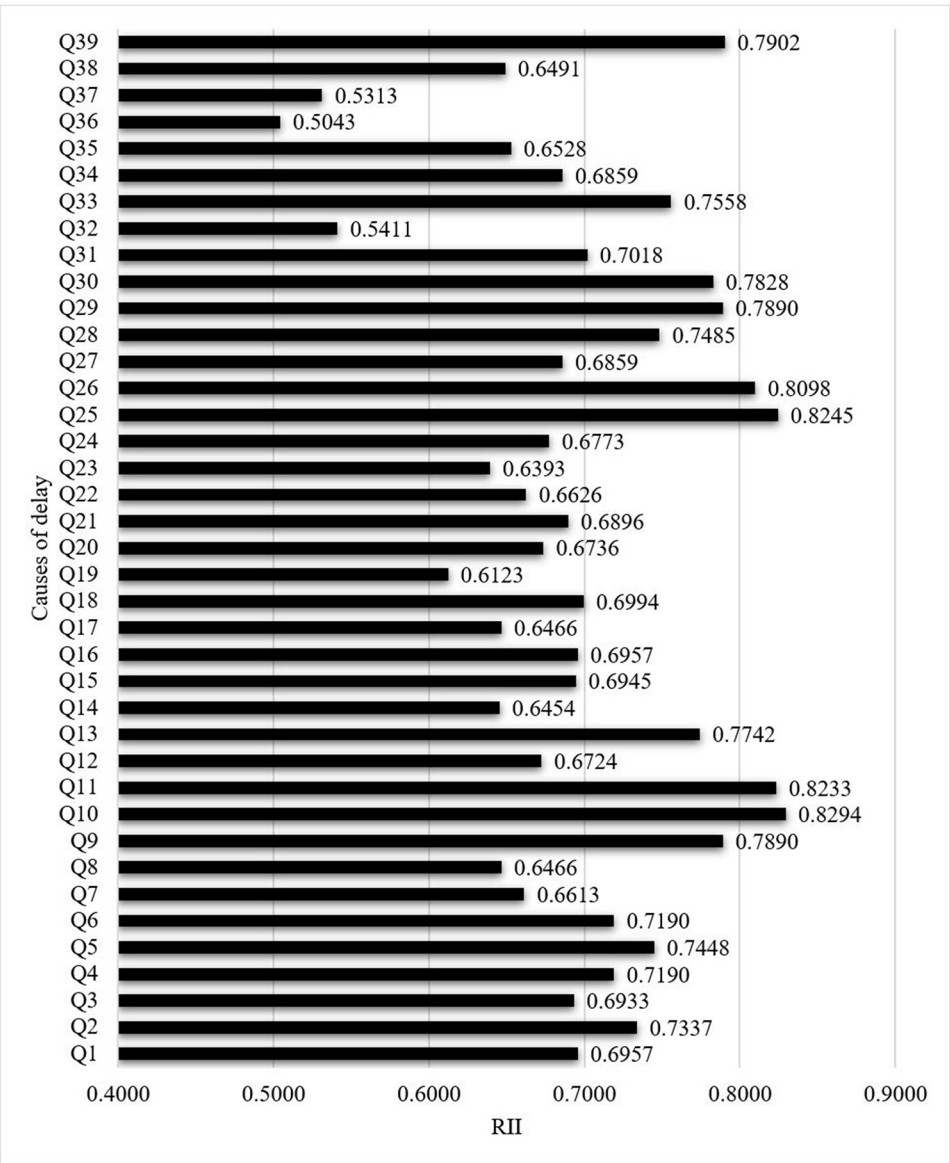

**Fig 3. RII vs causes of delay.** Source: Authors' illustration.

**Client/owner-related factors.** Table 5 shows the calculated RII and the ranking for eight factors categorised under client/owner-related factors.

Financial difficulties of the client were identified as the most significant factor in this group. The second highest ranked factor was the slowness of the client's decision-making. Delay in settling contractor claims by clients and delay in design approvals were the third most significant factors within the group. During the covid pandemic, claims from the government to contractors were delayed, as priority was given to healthcare and not development. According to the responses, errors in design and specifications were the least significant factors in delaying a construction project. This can be partly attributable to a majority of poorly unqualified subcontractors.

**Contractor-related factors.** The calculated RII and the ranking for eight factors categorised under contractor-related factors are shown in Table 6.

**Table 3. Mean value, standard deviation, RII and rank of each factor based on the type of respondent.**

| No | Client/Owner | | | | Contractor | | | | Consultant | | | |
|---|---|---|---|---|---|---|---|---|---|---|---|---|
| | Mean | SD | RII | Rank | Mean | SD | RII | Rank | Mean | SD | RII | Rank |
| Q1 | 3.1915 | 1.1912 | 0.6383 | 30 | 3.7391 | 1.2006 | 0.7478 | 16 | 3.4615 | 1.0012 | 0.6923 | 19 |
| Q2 | 3.2979 | 1.3008 | 0.6596 | 23 | 3.9348 | 0.9522 | 0.7870 | 8 | 3.7385 | 0.9886 | 0.7477 | 12 |
| Q3 | 3.0213 | 1.2596 | 0.6043 | 34 | 3.6957 | 0.9397 | 0.7391 | 17 | 3.5846 | 1.1577 | 0.7169 | 15 |
| Q4 | 3.4894 | 1.1396 | 0.6979 | 14 | 3.5435 | 1.0895 | 0.7087 | 21 | 3.6462 | 0.9426 | 0.7292 | 14 |
| Q5 | 3.5319 | 1.2132 | 0.7064 | 12 | 3.6739 | 1.2121 | 0.7348 | 19 | 3.8615 | 1.1842 | 0.7723 | 7 |
| Q6 | 3.4255 | 1.2810 | 0.6851 | 15 | 3.7826 | 1.0937 | 0.7565 | 12 | 3.5385 | 1.1467 | 0.7077 | 16 |
| Q7 | 3.1915 | 1.3292 | 0.6383 | 30 | 3.4130 | 1.0868 | 0.6826 | 27 | 3.2923 | 0.9474 | 0.6585 | 27 |
| Q8 | 3.0000 | 1.3831 | 0.6000 | 35 | 3.3478 | 1.0795 | 0.6696 | 31 | 3.2769 | 1.3051 | 0.6554 | 29 |
| Q9 | 3.9362 | 1.0715 | 0.7872 | 6 | 4.0652 | 0.9978 | 0.8130 | 4 | 3.8769 | 1.0681 | 0.7754 | 6 |
| Q10 | 4.1702 | 0.9628 | 0.8340 | 2 | 4.2174 | 0.9869 | 0.8435 | 1 | 4.1385 | 0.7474 | 0.8277 | 2 |
| Q11 | 4.1915 | 0.8246 | 0.8383 | 1 | 3.9565 | 1.0532 | 0.7913 | 7 | 4.1538 | 0.7548 | 0.8308 | 1 |
| Q12 | 3.3830 | 0.9453 | 0.6766 | 18 | 3.3478 | 0.8748 | 0.6696 | 31 | 3.2923 | 0.9638 | 0.6585 | 27 |
| Q13 | 4.0213 | 1.0932 | 0.8043 | 4 | 3.7609 | 1.1583 | 0.7522 | 14 | 3.8615 | 0.9663 | 0.7723 | 7 |
| Q14 | 3.2128 | 1.1409 | 0.6426 | 27 | 3.3043 | 1.0513 | 0.6609 | 34 | 3.1692 | 1.1260 | 0.6338 | 32 |
| Q15 | 3.2979 | 1.1963 | 0.6596 | 23 | 3.5435 | 1.0895 | 0.7087 | 21 | 3.5231 | 0.9860 | 0.7046 | 17 |
| Q16 | 3.5106 | 1.1396 | 0.7021 | 13 | 3.4783 | 1.0053 | 0.6957 | 24 | 3.4769 | 1.0017 | 0.6954 | 18 |
| Q17 | 3.1915 | 1.1159 | 0.6383 | 30 | 3.4783 | 1.0486 | 0.6957 | 24 | 3.0615 | 0.9663 | 0.6123 | 36 |
| Q18 | 3.3830 | 1.0745 | 0.6766 | 18 | 3.8043 | 1.0246 | 0.7609 | 11 | 3.3231 | 1.1471 | 0.6646 | 24 |
| Q19 | 2.7660 | 1.0046 | 0.5532 | 36 | 3.1957 | 1.1474 | 0.6391 | 36 | 3.1538 | 0.9558 | 0.6308 | 35 |
| Q20 | 3.2128 | 1.3821 | 0.6426 | 27 | 3.6739 | 1.2121 | 0.7348 | 19 | 3.2462 | 1.2504 | 0.6492 | 31 |
| Q21 | 3.3404 | 1.2385 | 0.6681 | 22 | 3.6957 | 1.0300 | 0.7391 | 17 | 3.3231 | 1.1740 | 0.6646 | 24 |
| Q22 | 3.2553 | 1.2592 | 0.6511 | 25 | 3.3913 | 1.1250 | 0.6783 | 29 | 3.3077 | 1.2365 | 0.6615 | 26 |
| Q23 | 3.0638 | 1.0916 | 0.6128 | 33 | 3.3696 | 0.9512 | 0.6739 | 30 | 3.1692 | 1.2320 | 0.6338 | 32 |
| Q24 | 3.4043 | 1.2452 | 0.6809 | 16 | 3.3261 | 1.0761 | 0.6652 | 33 | 3.4154 | 1.1976 | 0.6831 | 20 |
| Q25 | 4.0426 | 1.1221 | 0.8085 | 3 | 4.1739 | 1.0812 | 0.8348 | 2 | 4.1231 | 0.8387 | 0.8246 | 3 |
| Q26 | 3.9787 | 0.9888 | 0.7957 | 5 | 4.0870 | 1.0072 | 0.8174 | 3 | 4.0308 | 0.9009 | 0.8062 | 5 |
| Q27 | 3.3617 | 1.1502 | 0.6723 | 21 | 3.4130 | 1.0662 | 0.6826 | 27 | 3.4154 | 0.9665 | 0.6831 | 20 |
| Q28 | 3.6596 | 1.1088 | 0.7319 | 11 | 3.8696 | 0.9800 | 0.7739 | 9 | 3.7077 | 0.9308 | 0.7415 | 13 |
| Q29 | 3.9362 | 1.1113 | 0.7872 | 6 | 4.0435 | 1.0101 | 0.8087 | 6 | 3.8615 | 1.0588 | 0.7723 | 7 |
| Q30 | 3.8723 | 0.9235 | 0.7745 | 8 | 4.0652 | 0.8794 | 0.8130 | 4 | 3.8462 | 0.9054 | 0.7692 | 11 |
| Q31 | 3.3830 | 1.0945 | 0.6766 | 18 | 3.7826 | 1.0091 | 0.7565 | 12 | 3.3846 | 1.0708 | 0.6769 | 22 |
| Q32 | 2.5957 | 1.1546 | 0.5191 | 37 | 2.7609 | 0.9472 | 0.5522 | 37 | 2.6769 | 1.0017 | 0.5354 | 38 |
| Q33 | 3.7021 | 1.1405 | 0.7404 | 10 | 3.7609 | 0.9930 | 0.7522 | 14 | 3.8615 | 0.9823 | 0.7723 | 7 |
| Q34 | 3.4043 | 0.9704 | 0.6809 | 16 | 3.5217 | 1.0053 | 0.7043 | 23 | 3.3846 | 0.9633 | 0.6769 | 22 |
| Q35 | 3.2128 | 0.9766 | 0.6426 | 27 | 3.4348 | 1.1861 | 0.6870 | 26 | 3.1692 | 0.8398 | 0.6338 | 32 |
| Q36 | 2.4894 | 1.1772 | 0.4979 | 38 | 2.5870 | 1.1071 | 0.5174 | 39 | 2.5077 | 1.0019 | 0.5015 | 39 |
| Q37 | 2.4681 | 1.1951 | 0.4936 | 39 | 2.7391 | 1.0839 | 0.5478 | 38 | 2.7077 | 1.0857 | 0.5415 | 37 |
| Q38 | 3.2340 | 1.1461 | 0.6468 | 26 | 3.2826 | 1.1863 | 0.6565 | 35 | 3.2615 | 1.1079 | 0.6523 | 30 |
| Q39 | 3.8723 | 1.0346 | 0.7745 | 8 | 3.8261 | 0.9731 | 0.7652 | 10 | 4.1077 | 0.8315 | 0.8215 | 4 |

Source: Authors' calculations.

According to the results, the shortage of skilled subcontractors/suppliers was the top ranked factor within the group. Financial difficulties of contractors were identified as the second most significant factor which affects the timely completion of a construction project. The inability of the client to fund the contractor properly, can trouble the contractors when carrying out the

**Table 4. Top ranked delay factors for each group of respondents.**

| Client/owner | Contractor | Consultant | Rank |
|---|---|---|---|
| Financial difficulties of contractors | Shortage of skilled subcontractors/suppliers | Financial difficulties of contractors | 1 |
| Shortage of skilled subcontractors/suppliers | Shortage of labourers. (Skilled, semi-skilled, unskilled) | Shortage of skilled subcontractors/suppliers | 2 |
| Shortage of labourers (Skilled, semi-skilled, unskilled) | Delay in delivering materials to the site | Shortage of labourers (Skilled, semi-skilled, unskilled) | 3 |
| Poor site management, monitoring, and control | Poor planning and scheduling | Covid-19 pandemic situation | 4 |
| Delay in delivering materials to the site | Inadequate numbers of equipment | Delay in delivering materials to the site | 5 |
| Poor planning and scheduling | Fluctuation of material prices in the market | Poor planning and scheduling | 6 |
| Fluctuation of material prices in the market | Financial difficulties of contractors | Financial difficulties of client | 7 |
| Inadequate numbers of equipment | Slowness of the client's decision-making | Poor site management, monitoring, and control | 8 |
| Covid-19 pandemic situation | Low productivity of labourers | Fluctuation of material prices in the market | 9 |
| Delay in obtaining permissions/approvals from government | Covid-19 pandemic situation | Delay in obtaining permissions/approvals from government | 10 |

Source: Authors' calculations.

construction work. The third top ranked factor was poor planning and scheduling. The contractors should be able to prioritise the critical activities when planning the project at the initial stage. Ignoring such critical activities will lead to delays in the construction project. Errors during construction were identified as the least significant factor within the group.

**Consultant-related factors.** Table 7 shows the ranking and the calculated RII for the factors categorised under consultant-related factors. This group contained eight factors that delay a construction project.

Delay in material and payment approval was the most significant factor that could delay a construction project's completion. The consultants should be able to reduce the amount of time they take for material and payment approvals, to avoid delays in construction work. The second highest priority factor for delay included preparing and approving drawings and design documents. Poor coordination and communication between consultants and other stakeholders were the third highest ranked delay factor. The group's least important factor was errors in contract documents.

**Resource-related factors.** The calculated RII and the ranking of factors categorised under resource-related factors are shown in Table 8. Eight factors were included in this group.

The top ranking factor was the shortage of labourers (Skilled, semi-skilled, unskilled). As a result of the Covid-19 pandemic situation, employing labourers for construction projects has

**Table 5. RII and ranking of client/owner-related factors.**

| | Delay factors | RII | Rank |
|---|---|---|---|
| Q1 | Changes in design by the client during construction. | 0.6957 | 5 |
| Q2 | Slowness of the client's decision-making. | 0.7337 | 2 |
| Q3 | Unreasonable project duration given by the client. | 0.6933 | 6 |
| Q4 | Delay in settling contractor claims by the client. | 0.7190 | 3 |
| Q5 | Financial difficulties of the client. | 0.7448 | 1 |
| Q6 | Delay in design approvals. | 0.7190 | 3 |
| Q7 | Poor communication with contracting parties. | 0.6613 | 7 |
| Q8 | Errors in design and specifications. | 0.6466 | 8 |

Source: Authors' calculations.

**Table 6. RII and ranking of contractor-related factors.**

|  | Delay factors | RII | Rank |
|---|---|---|---|
| Q9 | Poor planning and scheduling. | 0.7890 | 3 |
| Q10 | Shortage of skilled subcontractors/suppliers. | 0.8294 | 1 |
| Q11 | Financial difficulties of contractors. | 0.8233 | 2 |
| Q12 | Disagreements between the contractor and other parties. | 0.6724 | 7 |
| Q13 | Poor site management, monitoring, and control. | 0.7742 | 4 |
| Q14 | Errors during construction. | 0.6454 | 8 |
| Q15 | Underestimating the project duration. | 0.6945 | 6 |
| Q16 | Regular changes of the subcontractor's staff. | 0.6957 | 5 |

Source: Authors' calculations.

been difficult. Due to lockdowns and travel restrictions, labourers could not travel from their home areas to the construction sites; also, it is unfeasible to provide accommodation to a large number of labourers. Delay in delivering materials to the site was the second highest ranked delay factor within the group. The third most important factor was the fluctuation of material prices in the market, another result of the economic crisis due to Covid-19. Among the eight factors, personal disagreements between labourers were found to be the least important project delay factor.

**External factors.** Table 9 shows the ranking and the calculated RII for the external factors. This group contained seven different delay factors.

According to the results, the current Covid-19 pandemic has adversely affected the construction industry. Hence, the results showed the Covid-19 pandemic situation as the top-ranked delay factor within the external factors group. Delay in obtaining approvals from the government was the second most important factor, while unknown subsurface conditions. (Soil condition, water table etc.) were identified as the third top-ranked delay factor. Accidents during construction were the least significant project delay factor.

## Ranking based on a group of factors

It is a crucial step in this study to identify the most significant group of factors which could affect the timely completion of a construction project. Therefore, the RII for each group of factors was found by calculating the average RII of the factors within each group. Table 10 shows the calculated RII and the ranking for each group of factors.

**Table 7. RII and ranking of consultant-related factors.**

|  | Delay factors | RII | Rank |
|---|---|---|---|
| Q17 | Delay in inspections and completed work approvals. | 0.6466 | 6 |
| Q18 | Delay in material and payment approval. | 0.6994 | 1 |
| Q19 | Errors in contract documents. | 0.6123 | 8 |
| Q20 | Constant design changes by the consultant | 0.6736 | 4 |
| Q21 | Delay in preparing and approving drawings and design documents. | 0.6896 | 2 |
| Q22 | Lack of experienced consultants. | 0.6626 | 5 |
| Q23 | Errors in design documents. | 0.6393 | 7 |
| Q24 | Poor coordination and communication. | 0.6773 | 3 |

Source: Authors' calculations.

Table 8. RII and ranking of resource-related factors.

|  | Delay factors | RII | Rank |
|---|---|---|---|
| Q25 | Shortage of labourers (Skilled, semi-skilled, unskilled) | 0.8245 | 1 |
| Q26 | Delay in delivering materials to the site. | 0.8098 | 2 |
| Q27 | Poor material handling at the site. | 0.6859 | 7 |
| Q28 | Low productivity of labourers. | 0.7485 | 5 |
| Q29 | Fluctuation of material prices in the market. | 0.7890 | 3 |
| Q30 | Inadequate numbers of equipment. | 0.7828 | 4 |
| Q31 | Failure of equipment. | 0.7018 | 6 |
| Q32 | Personal disagreements between labourers. | 0.5411 | 8 |

Source: Authors' calculations.

According to Table 10, the highest ranked group of factors was the contractor-related factors group. The second most important group of factors was the resource-related factors group. The third and fourth-ranked groups were client/owner-related factors group and the consultant-related factors group, respectively. The group which contained the external factors had the least influence on the timely completion of construction projects. Fig 4 shows the graphical depiction of the RII for each group of factors.

## Qualitative database created from the open-ended question in the questionnaire

Of the 163 respondents, 38 answered the open-ended question where they could present their opinion on construction project delays. Fig 5 presents the word cloud, which was generated using this question.

According to Fig 5, most of the respondents used the word "contractor" when providing their answers. The importance of obtaining this result was that it depicted a direct relationship with the results obtained from the descriptive analysis of the study, where the most influential group of factors was the "contractor-related factors" group. Secondly, the term "inexperience" has been used to describe the lack of skilled labourers, contractors, subcontractors, and suppliers at a construction site. The respondents have widely used words such as "materials", and "insufficient". This can be reconfirmed according to the descriptive analysis, where the "resource-related factors" was the second-ranked group of factors. The respondents have also used the word " political ", where political influence when selecting contractors, political interference on construction work and the country's political instability significantly influenced the delays in construction projects. Therefore, descriptive analysis results can be validated.

Table 9. RII and ranking of external factors.

|  | Delay factors | RII | Rank |
|---|---|---|---|
| Q33 | Delay in obtaining permissions/approvals from the government. | 0.7558 | 2 |
| Q34 | Unknown subsurface conditions. (Soil condition, water table etc.) | 0.6859 | 3 |
| Q35 | Bad weather conditions. | 0.6528 | 4 |
| Q36 | Accidents during construction. | 0.5043 | 7 |
| Q37 | Changes in laws and regulations from the government. | 0.5313 | 6 |
| Q38 | Delay in utility services. (Electricity, water etc.) | 0.6491 | 5 |
| Q39 | Covid-19 pandemic situation | 0.7902 | 1 |

Source: Authors' calculations.

**Table 10. RII and ranking of group of factors.**

| Group of factors | RII | Rank |
|---|---|---|
| Client/Owner related factors | 0.7017 | 3 |
| Contractor related factors | 0.7405 | 1 |
| Consultant related factors | 0.6626 | 4 |
| Resource related factors | 0.7354 | 2 |
| External factors | 0.6528 | 5 |

Source: Authors' calculations.

## Conclusion

The present study explored factors which could affect the timely completion of construction projects in Sri Lanka. The diverse perspectives of clients, contractors and consultants were considered to analyse the most significant factors for delays.

The study's main objective was to identify the most critical factors that could affect the timely completion of construction projects in Sri Lanka. Thirty nine factors were identified through a comprehensive literature review and expert opinion. These factors were categorised into five groups namely, client/owner related factors, contractor related factors, consultant related factors, resource related factors and external factors. A questionnaire was developed incorporating questions relevant to these factors, which was effective in collecting data from the selected 163 Civil Engineers in Sri Lanka. The collected data were analysed using the RII. The factors were ranked accordingly to achieve the main objective of the study. The top ten project delaying factors were identified as, "Shortage of skilled subcontractors/suppliers", "Shortage of labourers (Skilled, semi-skilled, unskilled)", "Financial difficulties of contractors", "Delay of delivering materials to site", "Covid-19 pandemic situation", "Fluctuation of material prices in the market", "Poor planning and scheduling", "Inadequate numbers of equipment", "Poor site management, monitoring, and control" and "Delay in obtaining permissions/ approvals from government". According to the clients/owners and consultants, "Financial

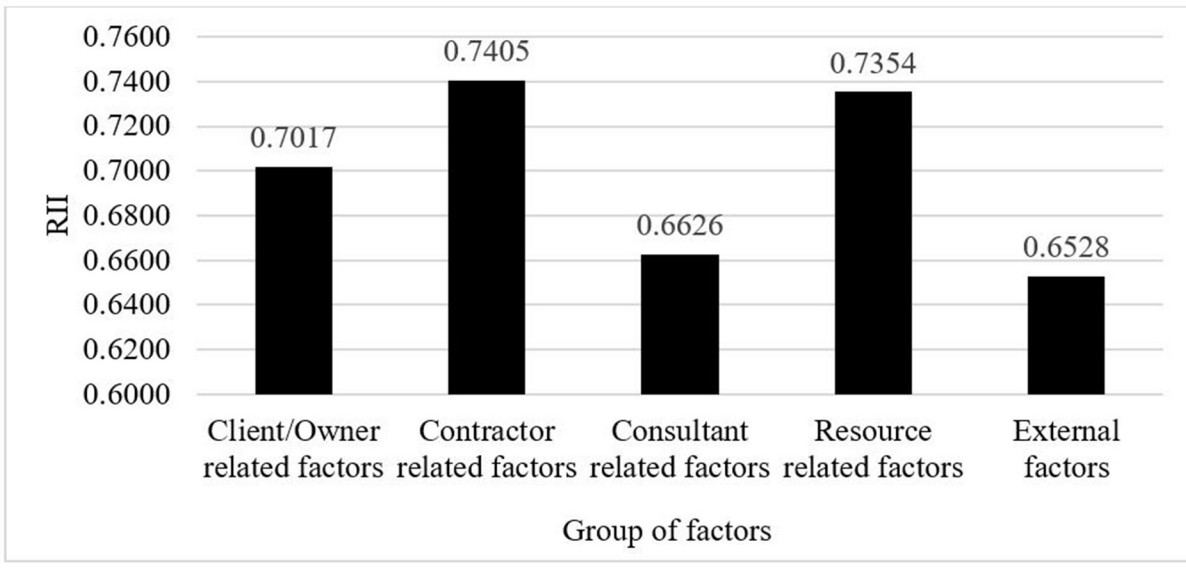

**Fig 4. Graphical depiction of RII for each group of factors.** Source: Authors' illustration.

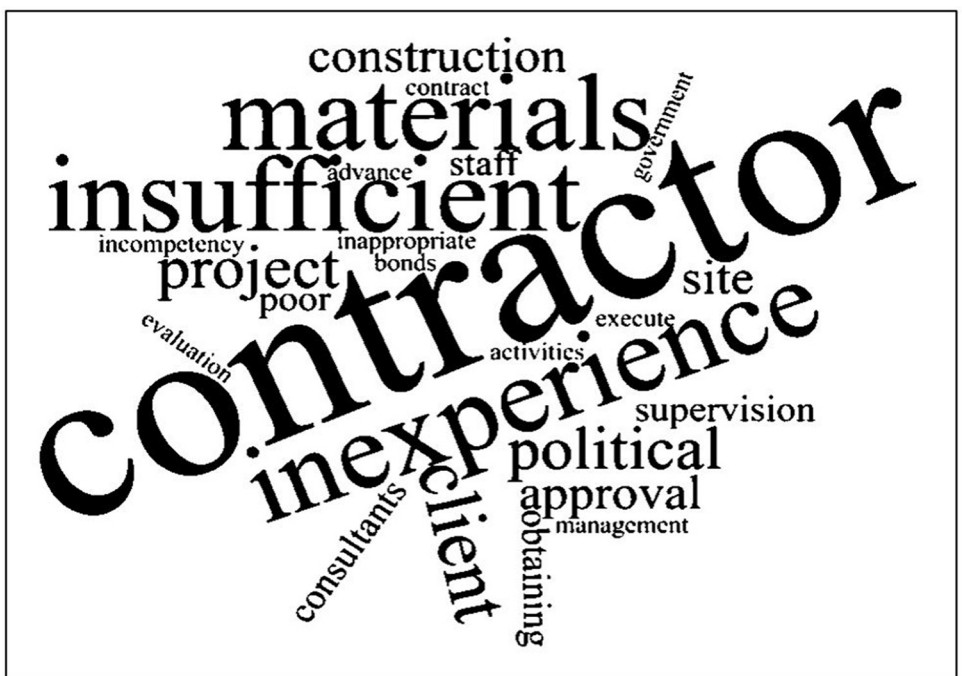

**Fig 5. Qualitative database.** Source: Generated by www.wordcloud.com.

difficulties of contractors" was the most influential factor which delays a construction project. The contractors claimed that the "Shortage of skilled subcontractors/suppliers" was the most significant project delaying factor. Also, the most important group of factors was identified as the contractor related factors group. This was further confirmed by the qualitative database obtained by the respondents' opinions. The most influential factor under the client/owner related factors was identified as the "Financial difficulties of clients". "Shortage of skilled subcontractors/suppliers" ranked at the top of contractor related factors. "Delay in material and payment approval" and "Shortage of labours (Skilled, semi-skilled, unskilled)" were the most significant factors under consultant related factors and resource related factors respectively. When it comes to the external factors, "Covid-19 pandemic situation" was identified as the top ranked project delaying factor. The study's findings could be used by the stakeholders of a construction project to avoid unnecessary delays.

The novelty of the study is that the insights would contribute to the engineering environment and are highly relevant to policymakers to improve the timely completion of construction projects in Sri Lanka. According to the present study's results, the lack of skilled subcontractors and suppliers and the lack of labourers are key issues to be addressed in timely completion of a construction project. Operating with inexperienced and unskilled subcontractors and suppliers may be challenging for the contractors to complete project tasks on time. Further, employing an adequate number of labourers is a must to avoid project delays. The contractors should be funded promptly by the owners so that the former can continue without interruptions to workflow, material shortages etc. As a result, the contractors could support the construction work as scheduled while considering the time and cost constraints. However, this could be much more difficult when the client/owner also faces financial difficulties. When the materials are not available at the construction site for critical activities, it will delay the construction work, whereas the labourers would idle with interruption to work. Therefore, facilitating the transportation of supplies to the construction site without any delays is mandatory.

This calls for effective logistics and supplies management. The material and payment processing (for timely approval etc.) should be done efficiently by the consultants to avoid any delays in the construction work. The construction industry suffered a setback amid the initial outbreak of the Covid-19 pandemic situation due to lockdowns and work disruptions. Moreover, labourers were scarce (as due to lockdown conditions, they had to be employed in their home areas); practical difficulties were observed when providing accommodation for the labourers (due to fear of Covid-19 health risks, renting was refused instead). Currently, the market price of materials has fluctuated considerably and is rising. This is because the pandemic has extended to an economic crisis, imports were restricted, and then supplies became limited and expensive. Thus, planning too has become challenging, adversely affecting the cost aspects, exceeding pro forma budgets in every construction project, which has affected its timely completion. The contractors should be able to properly plan and schedule the site work at the initial stage. Apart from these, the site work should be properly managed, monitored, and controlled by the contractor at the initiation stage. Rationally, if contractors are skilled and of integrity, their selection is independent and transparent, then time overruns can be minimised to a certain extent. The required construction equipment should be available in adequate numbers at the construction site. Providing efficient services in granting permissions and approvals by the government authorities plays a major role in completing a construction project within the given time, as it postpones the start of the project.

## Limitations

Even though the study contributes largely to the construction industry in Sri Lanka, there could be some limitations which can be addressed in future research. The sample size could be expanded to be representative, compared to the large population of Civil Engineers employed in different types of construction work in Sri Lanka. This approach can assist for wide coverage and a comprehensive study to gain useful findings to avoid construction delays and related losses.

## Recommendations and policy implications

Based on the study's results, the financial difficulties of both contractors and clients should be evaluated for the successful completion of construction projects. Also, experienced subcontractors, suppliers and labourers should be available to avoid project delays. A skilled workforce is a must but lacking in the Sri Lankan setting. A transparent approach and unbiased selection of suitable contractors are mandatory to minimise root causes.

The timely completion of any construction project has a significant effect on the success of the project. Currently, the Sri Lankan construction industry is undergoing a booming phase, meaning meeting high demand and avoiding interruptions are vital. Therefore, the construction projects must be delivered on time without any delays, as it can also affect the country's overall economy. Therefore, the results of this study will be useful for the stakeholders to identify areas that deserve much focus and attention. Based on the findings of this research, the stakeholders can plan adequately, schedule, control and monitor the construction activities. Further, they can avoid losing revenue, undue delays, incurring additional costs and ensure credibility. The Sri Lankan government can use the findings of this study to foresee the reasons for delays in various construction projects and devise necessary mitigation measures.

## Supporting information

**S1 Appendix. Questionnaire.**
(DOCX)

**S2 Appendix. Data file.**
(XLSX)

## Acknowledgments

The authors would like to thank Ms. Gayendri Karunarathne for proof-reading and editing this manuscript.

## Author Contributions

**Conceptualization:** Nadeesha Abeysinghe, Ruwan Jayathilaka.

**Data curation:** Nadeesha Abeysinghe.

**Formal analysis:** Nadeesha Abeysinghe.

**Methodology:** Nadeesha Abeysinghe.

**Project administration:** Ruwan Jayathilaka.

**Software:** Nadeesha Abeysinghe.

**Supervision:** Ruwan Jayathilaka.

**Validation:** Nadeesha Abeysinghe, Ruwan Jayathilaka.

**Visualization:** Nadeesha Abeysinghe.

**Writing – original draft:** Nadeesha Abeysinghe, Ruwan Jayathilaka.

**Writing – review & editing:** Ruwan Jayathilaka.

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
