## [Decision Letter · Decision Letter 0]

2 Sep 2022

PONE-D-22-23136Why projects delay?: A study of timely completion of construction projects in Sri LankaPLOS ONE

Dear Dr. Ruwan Jayathilaka,

Thank you for submitting your manuscript to PLOS ONE. After careful consideration, we feel that it has merit but does not fully meet PLOS ONE’s publication criteria as it currently stands. Therefore, we invite you to submit a revised version of the manuscript that addresses the points raised during the review process.

We look forward to receiving your revised manuscript.

Kind regards,

Dr. Anu Sayal, Ph.D.

Academic Editor

PLOS ONE

Reviewers' comments:

Reviewer's Responses to Questions

**Comments to the Author**

1. Is the manuscript technically sound, and do the data support the conclusions?

Reviewer #1: Partly

Reviewer #2: No

2. Has the statistical analysis been performed appropriately and rigorously? 

Reviewer #1: Yes

Reviewer #2: No

3. Have the authors made all data underlying the findings in their manuscript fully available?

Reviewer #1: Yes

Reviewer #2: Yes

4. Is the manuscript presented in an intelligible fashion and written in standard English?

Reviewer #1: No

Reviewer #2: No

5. Review Comments to the Author

Reviewer #1: This paper has attempted to: A study of timely completion of construction projects in Sri Lanka

The following items are helpful comments for the authors to consider:

• Please rewrite the title by using professional way.

• Please rewrite the abstract by using professional way.

• Use more technical keywords.

• The literature review is very shallow with no in-depth review for the previous related work and is not clear; I believe this could be significantly modified to give the important information.

• Problem description is not clear; I believe this should be significantly written in a more professional way.

Writing style of the manuscript is needed to be improved.

• Check the wiring sequence in the manuscript and improve it.

• Focus on your novelty and contribution to your paper especially at the conclusion.

• Discuss the formulas in more detail.

• Use the illustrative flowchart for methodology and sequence.

• More focus on questionnaire is needed.

• Please add all formulas with more details at your manuscript.

• Conclusion should be written in a more professional way.

• Please put list of abbreviations and symbols at the end of the manuscript.

• Some editorial and typo errors.

• There are some grammatical errors that need to be fixed.

• Please add and increase updated References with formatted outline.

Reviewer #2: The manuscript has been poorly written, while there many flaws. First of all, I have never seen any article which has an introduction section (problem statement) but without any references cited. This should mean that the entire section has been written based on your own findings.

Literature search has many missing articles on the subject and I don't understand the rationale behind the separation of the studies according to different countries, as this study is not a review study.

Literature review section is too long but not comprehensive.

6. PLOS authors have the option to publish the peer review history of their article (what does this mean?). If published, this will include your full peer review and any attached files.

Reviewer #1: No

Reviewer #2: No

---

## [Author Response · Author response to Decision Letter 0]

28 Sep 2022

Point by point response to editor and reviewers

Dear editor and the reviewers,

We would like to express our profound appreciation to the editor and the reviewers for the valuable comments and suggestions made on our manuscript which were very helpful in revising and improving it.

Please note that the line numbers referred in this document is aligned with the revised manuscript which has track changes.

Reviewer 1 comment 1: Please rewrite the title by using professional way.

Authors’ Response to Reviewer 1 Comment 1: Comment has been well noted and the title is rewritten using professional way. This has included from line 1 to 2 and 26 to 27 as follows, 

“A study on timely completion of construction projects in Sri Lanka”

Reviewer 1 comment 2: Please rewrite the abstract by using professional way.

Authors’ Response to Reviewer 1 Comment 2: Comment has been noted and the abstract is rewritten using professional way. This has included from line 29 to 47 as follows, 

“Timely completion is a crucial factor for the success of a construction project, especially in the Sri Lankan context. This study aims to identify the most influential factors that affect the timely completion of construction projects in Sri Lanka. Thirty-nine factors were identified through a comprehensive literature review and experts’ opinions. A questionnaire was distributed incorporating the 39 project delay factors, and responses from 163 Civil Engineers were obtained. The Relative Importance Index (RII) analysed and ranked the project delay factors. The top ranked significant project delay factors were identified as shortage of skilled subcontractors/suppliers, shortage of labourers (Skilled, semi-skilled, unskilled), financial difficulties of contractors, delay in delivering materials to the site, and Covid-19 pandemic situation. According to the main three respondent types, i.e., clients/owners, contractors and consultants, the contractor related factors was the key group among others that delay a construction project. The scientific value of the study includes assisting the Sri Lankan construction industry to identify the factors affecting the timely completion of construction projects, and developing mitigation methods and strategies. Also, the stakeholders could duly schedule the construction work by identifying areas that need more attention. The contribution of this study would assist stakeholders to adopt a proactive approach by identifying mistakes on their part and minimising potential issues that lead to construction project delays in Sri Lanka.”

Reviewer 1 comment 3: Use more technical keywords.

Authors’ Response to Reviewer 1 Comment 3: Comment has been noted and it has been corrected in the revised manuscript from line number 49 to 50.

Reviewer 1 comment 4: The literature review is very shallow with no in-depth review for the previous related work and is not clear; I believe this could be significantly modified to give the important information.

Authors’ Response to Reviewer 1 Comment 4: Comment has been well noted and the literature review was modified by providing more information. This has corrected from line 130 to 297.

Reviewer 1 comment 5: Problem description is not clear; I believe this should be significantly written in a more professional way. Writing style of the manuscript is needed to be improved.

Authors’ Response to Reviewer 1 Comment 5: Comment has been well noted. The problem statement is included in the revised manuscript from line 67 to line 93.

Reviewer 1 comment 6: Check the wiring sequence in the manuscript and improve it.

Authors’ Response to Reviewer 1 Comment 6: Comment has been well noted. The writing of the manuscript is improved using professional way. The manuscript was also proofread and edited by a proof-reader.

Reviewer 1 comment 7: Focus on your novelty and contribution to your paper especially at the conclusion.

Authors’ Response to Reviewer 1 Comment 7: Thank you. The comment has been noted. The novelty and contribution of the study is included from line 581 to line 605 as follows,

“The novelty of the study is that the insights would contribute to the engineering environment and are highly relevant to policymakers to improve the timely completion of construction projects in Sri Lanka. According to the present study’s results, the lack of skilled subcontractors and suppliers and the lack of labourers are key issues to be addressed in timely completion of a construction project. Operating with inexperienced and unskilled subcontractors and suppliers may be challenging for the contractors to complete project tasks on time. Further, employing an adequate number of labourers is a must to avoid project delays. The contractors should be funded promptly by the owners so that the former can continue without interruptions to workflow, material shortages etc. As a result, the contractors could support the construction work as scheduled while considering the time and cost constraints. However, this could be much more difficult when the client/owner also faces financial difficulties. When the materials are not available at the construction site for critical activities, it will delay the construction work, whereas the labourers would idle with interruption to work. Therefore, facilitating the transportation of supplies to the construction site without any delays is mandatory. This calls for effective logistics and supplies management. The material and payment processing (for timely approval etc.) should be done efficiently by the consultants to avoid any delays in the construction work. The construction industry suffered a setback amid the initial outbreak of the covid-19 pandemic situation due to lockdowns and work disruptions. Moreover, labourers were scarce (as due to lockdown conditions, they had to be employed in their home areas); practical difficulties were observed when providing accommodation for the labourers (due to fear of Covid-19 health risks, renting was refused instead).”

Reviewer 1 comment 8: Discuss the formulas in more detail.

Authors’ Response to Reviewer 1 Comment 8: Comment has been well noted. The formula which was used in the study was strengthen by adding following paragraph. Please see lines from 356 to line 362.

“Weights were assigned to each factor by each respondent (1= Very low significance, 2= Low significance, 3= Average significance, 4= High significance, 5= Very high significance) were multiplied by the frequency of responses given to each factor, and the total sum of those two values was calculated. The result was divided by the multiplication of the highest weight (5) and the total count of respondents. The ranking of the delay factors was done using the RII. Overall rankings were calculated by combining the responses given by all the respondents.”

Reviewer 1 comment 9: Use the illustrative flowchart for methodology and sequence.

Authors’ Response to Reviewer 1 Comment 9: Comment has been noted and Figure 2 was added to the revised manuscript to show flowchart for methodology and research sequence. This is sated in line number 306 to 307.

Reviewer 1 comment 10: More focus on questionnaire is needed.

Authors’ Response to Reviewer 1 Comment 10: Comment has been well noted. Detail discussion of the questionnaire has been added to the revised manuscript. This has included from line 318 to line 330.

Reviewer 1 comment 11: Please add all formulas with more details at your manuscript.

Authors’ Response to Reviewer 1 Comment 11: Comment has been well noted. The one formula which was used in the study was discussed in more detail from line 347 to line 362.

Reviewer 1 comment 12: Conclusion should be written in a more professional way.

Authors’ Response to Reviewer 1 Comment 12: Comment has been noted and the conclusion is rewritten in a more professional way. This has included from line 570 to 526.

Reviewer 1 comment 13: Please put list of abbreviations and symbols at the end of the manuscript.

Authors’ Response to Reviewer 1 Comment 13: Thank you for the comment. As per the PLOS ONE journal guideline abbreviation should define upon the first appearance in the text. Not as the list of abbreviation. Please check the following link for more details.

https://journals.plos.org/plosone/s/submission-guidelines#loc-abbreviations

Authors would like to keep the abbreviation as per the PLOS ONE guidelines.

Reviewer 1 comment 14: Some editorial and typo errors.

Authors’ Response to Reviewer 1 Comment 14: Comment has been noted. The editorial and typo errors were corrected in the revised manuscripts.

Reviewer 1 comment 15: There are some grammatical errors that need to be fixed.

Authors’ Response to Reviewer 1 Comment 15: The revised manuscript was also proofread once the changes were done and therefore, now, we can confirm that the revised manuscript is free of punctuation and grammatical errors.

Reviewer 1 comment 16: Please add and increase updated References with formatted outline.

Authors’ Response to Reviewer 1 Comment 16: Comment has been well noted and the references has been updated in revised manuscript.

Reviewer 2 comment 1: The manuscript has been poorly written, while there many flaws. First of all, I have never seen any article which has an introduction section (problem statement) but without any references cited. This should mean that the entire section has been written based on your own findings.

Authors’ Response to Reviewer 2 Comment 1: Thank you. Comment has been well noted. The citations were updated in the introduction section, and the problem statement is also included in the revised manuscript from line 67 to line 93.

Reviewer 2 comment 2: Literature search has many missing articles on the subject and I don't understand the rationale behind the separation of the studies according to different countries, as this study is not a review study.

Authors’ Response to Reviewer 2 Comment 2: Comment has been noted. The reason for separating the studies based on the continents was included from line 133 to line 138.

Reviewer 2 comment 3: Literature review section is too long but not comprehensive.

Authors’ Response to Reviewer 2 Comment 3: Thank you. Comment has been well noted and the literature review was modified by providing important information in the revised manuscript from line number 130 to 297.

---

## [Decision Letter · Decision Letter 1]

21 Oct 2022

PONE-D-22-23136R1A study on timely completion of construction projects in Sri LankaPLOS ONE

Dear Dr. Ruwan Jayathilaka,

Thank you for submitting your manuscript to PLOS ONE. After careful consideration, we feel that it has merit but does not fully meet PLOS ONE’s publication criteria as it currently stands. Therefore, we invite you to submit a revised version of the manuscript that addresses the points raised during the review process.

We look forward to receiving your revised manuscript.

Kind regards,

Dr. Anu Sayal, Ph.D.

Academic Editor

PLOS ONE

Journal Requirements:

Reviewers' comments:

Reviewer's Responses to Questions

**Comments to the Author**

1. If the authors have adequately addressed your comments raised in a previous round of review and you feel that this manuscript is now acceptable for publication, you may indicate that here to bypass the “Comments to the Author” section, enter your conflict of interest statement in the “Confidential to Editor” section, and submit your "Accept" recommendation.

Reviewer #1: All comments have been addressed

Reviewer #3: (No Response)

2. Is the manuscript technically sound, and do the data support the conclusions?

Reviewer #1: Yes

Reviewer #3: Yes

3. Has the statistical analysis been performed appropriately and rigorously? 

Reviewer #1: Yes

Reviewer #3: Yes

4. Have the authors made all data underlying the findings in their manuscript fully available?

Reviewer #1: Yes

Reviewer #3: Yes

5. Is the manuscript presented in an intelligible fashion and written in standard English?

Reviewer #1: Yes

Reviewer #3: Yes

6. Review Comments to the Author

Reviewer #1: (No Response)

Reviewer #3: Please support the efforts already put in this work by making further revision on the areas highighted in the review report

7. PLOS authors have the option to publish the peer review history of their article (what does this mean?). If published, this will include your full peer review and any attached files.

Reviewer #1: No

Reviewer #3: No

---

## [Author Response · Author response to Decision Letter 1]

29 Oct 2022

Point by point response to editor and reviewers

Dear editor and the reviewers,

We would like to express our profound appreciation to the editor and the reviewers for the valuable comments and suggestions made on our manuscript which were very helpful in revising and improving it.

Please note that the line numbers referred in this document is aligned with the revised manuscript which has track changes.

Reviewer’s Comment 1: Suggested improvement in Title: Factors influencing the timely completion of construction projects in Sri Lanka

Authors’ Response to Comment 1: Thank you. Comment has been well noted and the suggested title has been used for as the new title of the revised manuscript from line 2 to 3 and 28 to 29.

“Factors influencing the timely completion of construction projects in Sri Lanka”

Reviewer’s Comment 2: Sampling technique used was not captured in the abstract.

Authors’ Response to Comment 2: Comment has been well noted. The sampling technique which was used in the has been included in the revised manuscript line number 36 to 37.

“A questionnaire incorporating the 39 project delay factors was distributed among 163 Civil Engineers, and responses were obtained. Random sampling method was adopted to select the sample.”

Reviewer’s Comment 3: Improve the keywords, see suggestions; Construction projects, Time overrun, Relative Importance Index (RII), Delay factors,

Sri Lanka

Authors’ Response to Comment 3: Comment has been well noted. The suggested keywords have been added to the revised manuscript line number from 51 to 52.

“Keywords: Construction projects, Time Overrun, Relative Importance Index (RII), Delay factors, Sri Lanka”

Reviewer’s Comment 4: There are a lot of grammar errors and poor sentence construction. Avoid short sentences that do not make a complete sense.

Authors’ Response to Comment 4: Comment has been well noted. Grammar errors and poor sentence construction have been resolved in the revised manuscript.

Reviewer’s Comment 5: The introduction section says a lot about your work. You have to strengthen the research problem and gap. Consult more literature, there a lot of them on time performances factors in construction. Just 7 citations in this very important subject are not enough. Remove less important sentences; be concise in your approach.

Authors’ Response to Comment 5: Comment has been well noted. More literature have been added in the revised manuscript from line 68 to 84.

“The nature of the construction industry can be considered as uncertain. Construction projects differ from each other depending on the project size, project objectives, project duration, etc. Every project is unique on its own and no project has the same characteristics. Even though the construction projects nowadays use advanced and new project management theories and technologies, the delay in the completion of projects cannot be mitigated [3]. 

The time deviation of a construction project can be defined as the difference between specified project duration and the real project duration. There can be three types of time deviations in a construction project. Firstly, is a negative deviation, where the real duration is less than the specified duration. Secondly, there is the no particular deviation type, where the specified duration and the real duration are the same. Thirdly, is the positive deviation, where the real duration is greater than the specified duration. This positive deviation is also known as the time overrun, where the delays in the project completions occur. When the delay period is long, consequently the effects will also be greater/significant, which can exert a negative impact on the project. For the successful completion of a project, cost, quality as well as time, should be properly utilised [4].”

Reviewer’s Comment 6: The review section is satisfactory. However, some grammar editing is required.

Authors’ Response to Comment 6: Comment has been well noted. Grammar editing has been done for the revised manuscript.

Reviewer’s Comment 7: Why only civil engineers: Justify please. Could you not have obtained robust data if other experts in the construction industry were included?

Authors’ Response to Comment 7: Comment has been well noted. The reason for obtaining data from Civil Engineer has been included in the revised manuscript line number from 323 to 325.

“….Qualified Civil Engineers were selected as the population, since they are engaged in the construction work under all three stakeholder groups namely, client/owner, consultant and contractor…..”

Reviewer’s Comment 8: Sampling method adopted in the electronic administration of questionnaire was not stated.

Authors’ Response to Comment 8: Comment has been well noted. The sampling method is included in line 328 in the revised manuscript.

“…Then, the final questionnaire was distributed to 1,416 respondents selected using a random sampling method from 2,716 Civil Engineers included in the list…..”

Reviewer’s Comment 9: Justify the choice of questionnaire. Why was questionnaire used in this study?

Authors’ Response to Comment 9: Comment has been well noted. The reason for selecting questionnaire method has been include from line 302 to 304 in the revised manuscript.

“As the research strategy, survey strategy was adopted for the study where a questionnaire was developed in which the project delay factors were included. The respondents were able to rank each factor according to their significance, with the use of a questionnaire….”

Reviewer’s Comment 10: There was the lack of a clear explanation/description of the position of the outcome of this study with existing knowledge in the same subject area. Relate your results/finding to existing reports on the subject area. Please discussion your findings. How does it relate to the review you did?

Authors’ Response to Comment 10: Comment has been well noted. This has been included in the revised manuscript from line 389 to 400.

“…This result alligned with the results obtained by Yap, Goay (11) and Wang, Ford (22). Shortage of labourers (Skilled, semi-skilled, unskilled) was the second most significant factor (RII=0.8245) categorised under resource-related factors. This result can be further confirmed by the results obtained by Durdyev, Omarov (6) who conducted a study in Cambodia to identify the time overrun factors in residential building construction projects in the country. These factors seem realistic in the construction industry as most subcontracts/suppliers and labourers are not skilled. The third highest ranked factor was the financial difficulties of contractors (RII=0.8233), another contractor-related factor. In Malaysia, two research studies (9), (11) confirmed that the financial difficulties of the contractor play a major role in delaying construction projects, while Prasad, Vasugi (23) have also received the same result in the research they conducted in India…..”

Reviewer’s Comment 11: Please review this section to showcase and improve on the efforts put in previous sections. Especially regarding key factors and relationships. Avoid unnecessary ingredients in the conclusion section.

Authors’ Response to Comment 11: Comment has been well noted. This has been included in the revised manuscript from line 561 to 584.

“…..These factors were categorised into five groups namely, client/owner related factors, contractor related factors, consultant related factors, resource related factors and external factors. A questionnaire was developed incorporating questions relevant to these factors, which was effective in collecting data from the selected 163 Civil Engineers in Sri Lanka. The collected data were analysed using the RII. The factors were ranked accordingly to achieve the main objective of the study. The top ten project delaying factors were identified as, “Shortage of skilled subcontractors/suppliers”, “Shortage of labourers (Skilled, semi-skilled, unskilled)”, “Financial difficulties of contractors”, “Delay of delivering materials to site”, “Covid-19 pandemic situation”, “Fluctuation of material prices in the market”, “Poor planning and scheduling”, “Inadequate numbers of equipment”, “Poor site management, monitoring, and control” and “Delay in obtaining permissions/approvals from government”. According to the clients/owners and consultants, “Financial difficulties of contractors” was the most influential factor which delays a construction project. The contractors claimed that the “Shortage of skilled subcontractors/suppliers” was the most significant project delaying factor. Also, the most important group of factors was identified as the contractor related factors group. This was further confirmed by the qualitative database obtained by the respondents’ opinions. The most influential factor under the client/owner related factors was identified as the “Financial difficulties of clients”. “Shortage of skilled subcontractors/suppliers” ranked at the top of contractor related factors. “Delay in material and payment approval” and “Shortage of labours (Skilled, semi-skilled, unskilled)” were the most significant factors under consultant related factors and resource related factors respectively. When it comes to the external factors, “Covid-19 pandemic situation” was identified as the top ranked project delaying factor…..”

Reviewer’s Comment 12: Does this study have limitations? What recommendation do you have for future studies?

Authors’ Response to Comment 12: Comment has been well noted. Limitations of the study has been included in the revised manuscript from line 620 to 626. Recommendations has been included from line 628 to 643.

“Even though the study contributes largely to the construction industry in Sri Lanka, there could be some limitations which can be addressed in future research. The sample size could be expanded to be representative, compared to the large population of Civil Engineers employed in different types of construction work in Sri Lanka. This approach can assist for wide coverage and a comprehensive study to gain useful findings to avoid construction delays and related losses.”

“Based on the study's results, the financial difficulties of both contractors and clients should be evaluated for the successful completion of construction projects. Also, experienced subcontractors, suppliers and labourers should be available to avoid project delays. A skilled workforce is a must but lacking in the Sri Lankan setting. A transparent approach and unbiased selection of suitable contractors are mandatory to minimise root causes.

The timely completion of any construction project has a significant effect on the success of the project. Currently, the Sri Lankan construction industry is undergoing a booming phase, meaning meeting high demand and avoiding interruptions are vital. Therefore, the construction projects must be delivered on time without any delays, as it can also affect the country's overall economy. Therefore, the results of this study will be useful for the stakeholders to identify areas that deserve much focus and attention. Based on the findings of this research, the stakeholders can plan adequately, schedule, control and monitor the construction activities. Further, they can avoid losing revenue, undue delays, incurring additional costs and ensure credibility. The Sri Lankan government can use the findings of this study to foresee the reasons for delays in various construction projects and devise necessary mitigation measures.”

Reviewer’s Comment 13: Does this study have implications?

Authors’ Response to Comment 13: Comment has been well noted. Policy implications have been added in the revised manuscript from line 629 to 643. 

“Based on the study's results, the financial difficulties of both contractors and clients should be evaluated for the successful completion of construction projects. Also, experienced subcontractors, suppliers and labourers should be available to avoid project delays. A skilled workforce is a must but lacking in the Sri Lankan setting. A transparent approach and unbiased selection of suitable contractors are mandatory to minimise root causes.

The timely completion of any construction project has a significant effect on the success of the project. Currently, the Sri Lankan construction industry is undergoing a booming phase, meaning meeting high demand and avoiding interruptions are vital. Therefore, the construction projects must be delivered on time without any delays, as it can also affect the country's overall economy. Therefore, the results of this study will be useful for the stakeholders to identify areas that deserve much focus and attention. Based on the findings of this research, the stakeholders can plan adequately, schedule, control and monitor the construction activities. Further, they can avoid losing revenue, undue delays, incurring additional costs and ensure credibility. The Sri Lankan government can use the findings of this study to foresee the reasons for delays in various construction projects and devise necessary mitigation measures.”

Reviewer’s Comment 14: Revisit the reference list and make sure it conforms to the journals’ guide/format

Authors’ Response to Comment 14: Comment has been well noted. Reference list has been created as per the PLOS ONE journal guidelines. Please check the following link for more details. https://journals.plos.org/plosone/s/submission-guidelines/#loc-reference-style

---

## [Editor Report · Decision Letter 2]

4 Nov 2022

PONE-D-22-23136R2Factors influencing the timely completion of construction projects in Sri LankaPLOS ONE

Dear Dr. Jayathilaka,

Thank you for submitting your manuscript to PLOS ONE. After careful consideration, we feel that it has merit but does not fully meet PLOS ONE’s publication criteria as it currently stands. Therefore, we invite you to submit a revised version of the manuscript that addresses the points raised during the review process.

We look forward to receiving your revised manuscript.

Kind regards,

Anu Sayal, Ph.D.

Academic Editor

PLOS ONE

Journal Requirements:

Additional Editor Comments:

**Suggested improvement in Title**: Factors influencing the timely completion of construction projects in Sri Lanka

**ABSTRACT**
Sampling technique used was not captured in the abstract.

**KEYWORDS-improve the keywords, see suggestions**
Construction projects, Time overrun, Relative Importance Index (RII), Delay factors, Sri Lanka

**INTRODUCTION SECTION**

There a lot of grammar errors and poor sentence construction. Avoid short sentences that do not make a complete sense. There a lot of these type of sentences, for example,The introduction section says a lot about your work. You have to strengthen the research problem and gap. Consult more literature, there a lot of them on time performances factors in construction. Just 7 citations in this very important subject are not enough. Remove less important sentences; be concise in your approach.

**LITERATURE REVIEW SECTION**

 The review section is satisfactory, however, some grammar editing is required

**METHODOLOGY SECTION**

**Overall, the methodology adopted is satisfactory, bit it can be made better by attending to the followings;**

Why only civil engineers: Justify please. Could you not have obtained robust data if other experts in the construction industry were included?Sampling method adopted in the electronic administration of questionnaire was not stated.Justify the choice of questionnaire. Why was questionnaire used in this study?

**DATA ANALYSIS SECTION**

There was the lack of a clear explanation/description of the position of the outcome of this study with existing knowledge in the same subject area. Relate your results/finding to existing reports on the subject area. Please discussion your findings. How does it relate to the review you did?

**CONCLUSIONS AND RECOMMENDATIONS**

Please review this section to showcase and improve on the efforts put in previous sections. Especially regarding key factors and relationships. Avoid unnecessary ingredients in the conclusion section.Does this study have limitations? What recommendation do you have for future studies?Does this study have implications?

**REFERENCES**

Revisit the reference list and make sure it conforms to the journals’ guide/format
---

## [Author Response · Author response to Decision Letter 2]

5 Nov 2022

Point by point response to editor and reviewers

Dear editor and the reviewers,

We would like to express our profound appreciation to the editor and the reviewers for the valuable comments and suggestions made on our manuscript which were very helpful in revising and improving it.

Please note that the line numbers referred in this document is aligned with the revised manuscript which has track changes.

Comments of Editor

Journal Requirements: Please review your reference list to ensure that it is complete and correct. If you have cited papers that have been retracted, please include the rationale for doing so in the manuscript text, or remove these references and replace them with relevant current references. Any changes to the reference list should be mentioned in the rebuttal letter that accompanies your revised manuscript. If you need to cite a retracted article, indicate the article’s retracted status in the References list and also include a citation and full reference for the retraction notice.

Authors’ Response to Editor: Thank you very much and comment has been well noted. Reference list has been created as per the PLOS ONE journal guidelines. PLOS ONE uses “Vancouver” style and authors also double check the referencing style with the recently published articles in PLOS ONE. Authors would like to confirm reference list is complete and correct. As per the reviewer 1 feedback below, following two references have been added to the introduction section of the revised manuscript to strengthen the research problem and gap of the study.

3. Singh R. Delays and cost overruns in infrastructure projects: extent, causes and remedies. Economic and Political Weekly. 2010:43-54.

4. Catalão FP, Cruz CO, Sarmento JM. The determinants of time overruns in Portuguese public projects. Journal of Infrastructure Systems. 2021;27(2):05021002.

In our previous manuscript there were 41 references and it has increased to 43 with the current revised manuscript. Since this study is related to the Sri Lanka authors would like to keep the Sri Lankan literature. 

Addition Editor Comments:

Reviewer’s Comment 1: Suggested improvement in Title: Factors influencing the timely completion of construction projects in Sri Lanka

Authors’ Response to Comment 1: Thank you. Comment has been well noted and the suggested title has been used for as the new title of the revised manuscript from line 2 to 3 and 28 to 29.

“Factors influencing the timely completion of construction projects in Sri Lanka”

Reviewer’s Comment 2: Sampling technique used was not captured in the abstract.

Authors’ Response to Comment 2: Comment has been well noted. The sampling technique which was used in the has been included in the revised manuscript line number 36 to 37.

“A questionnaire incorporating the 39 project delay factors was distributed among 163 Civil Engineers, and responses were obtained. Random sampling method was adopted to select the sample.”

Reviewer’s Comment 3: Improve the keywords, see suggestions; Construction projects, Time overrun, Relative Importance Index (RII), Delay factors,

Sri Lanka

Authors’ Response to Comment 3: Comment has been well noted. The suggested keywords have been added to the revised manuscript line number from 51 to 52.

“Keywords: Construction projects, Time Overrun, Relative Importance Index (RII), Delay factors, Sri Lanka”

Reviewer’s Comment 4: There are a lot of grammar errors and poor sentence construction. Avoid short sentences that do not make a complete sense.

Authors’ Response to Comment 4: Comment has been well noted. Grammar errors and poor sentence construction have been resolved in the revised manuscript.

Reviewer’s Comment 5: The introduction section says a lot about your work. You have to strengthen the research problem and gap. Consult more literature, there a lot of them on time performances factors in construction. Just 7 citations in this very important subject are not enough. Remove less important sentences; be concise in your approach.

Authors’ Response to Comment 5: Comment has been well noted. More literature have been added in the revised manuscript from line 68 to 84. The following references have been added to revised manuscript to strengthen the research problem and the research gap.

3. Singh R. Delays and cost overruns in infrastructure projects: extent, causes and remedies. Economic and Political Weekly. 2010:43-54.

4. Catalão FP, Cruz CO, Sarmento JM. The determinants of time overruns in Portuguese public projects. Journal of Infrastructure Systems. 2021;27(2):05021002.

“The nature of the construction industry can be considered as uncertain. Construction projects differ from each other depending on the project size, project objectives, project duration, etc. Every project is unique on its own and no project has the same characteristics. Even though the construction projects nowadays use advanced and new project management theories and technologies, the delay in the completion of projects cannot be mitigated [3]. 

The time deviation of a construction project can be defined as the difference between specified project duration and the real project duration. There can be three types of time deviations in a construction project. Firstly, is a negative deviation, where the real duration is less than the specified duration. Secondly, there is the no particular deviation type, where the specified duration and the real duration are the same. Thirdly, is the positive deviation, where the real duration is greater than the specified duration. This positive deviation is also known as the time overrun, where the delays in the project completions occur. When the delay period is long, consequently the effects will also be greater/significant, which can exert a negative impact on the project. For the successful completion of a project, cost, quality as well as time, should be properly utilised [4].”

Reviewer’s Comment 6: The review section is satisfactory. However, some grammar editing is required.

Authors’ Response to Comment 6: Comment has been well noted. Grammar editing has been done for the revised manuscript.

Reviewer’s Comment 7: Why only civil engineers: Justify please. Could you not have obtained robust data if other experts in the construction industry were included?

Authors’ Response to Comment 7: Comment has been well noted. The reason for obtaining data from Civil Engineer has been included in the revised manuscript line number from 323 to 325.

“….Qualified Civil Engineers were selected as the population, since they are engaged in the construction work under all three stakeholder groups namely, client/owner, consultant and contractor…..”

Reviewer’s Comment 8: Sampling method adopted in the electronic administration of questionnaire was not stated.

Authors’ Response to Comment 8: Comment has been well noted. The sampling method is included in line 328 in the revised manuscript.

“…Then, the final questionnaire was distributed to 1,416 respondents selected using a random sampling method from 2,716 Civil Engineers included in the list…..”

Reviewer’s Comment 9: Justify the choice of questionnaire. Why was questionnaire used in this study?

Authors’ Response to Comment 9: Comment has been well noted. The reason for selecting questionnaire method has been include from line 302 to 304 in the revised manuscript.

“As the research strategy, survey strategy was adopted for the study where a questionnaire was developed in which the project delay factors were included. The respondents were able to rank each factor according to their significance, with the use of a questionnaire….”

Reviewer’s Comment 10: There was the lack of a clear explanation/description of the position of the outcome of this study with existing knowledge in the same subject area. Relate your results/finding to existing reports on the subject area. Please discussion your findings. How does it relate to the review you did?

Authors’ Response to Comment 10: Comment has been well noted. This has been included in the revised manuscript from line 389 to 400.

“…This result alligned with the results obtained by Yap, Goay (11) and Wang, Ford (22). Shortage of labourers (Skilled, semi-skilled, unskilled) was the second most significant factor (RII=0.8245) categorised under resource-related factors. This result can be further confirmed by the results obtained by Durdyev, Omarov (6) who conducted a study in Cambodia to identify the time overrun factors in residential building construction projects in the country. These factors seem realistic in the construction industry as most subcontracts/suppliers and labourers are not skilled. The third highest ranked factor was the financial difficulties of contractors (RII=0.8233), another contractor-related factor. In Malaysia, two research studies (9), (11) confirmed that the financial difficulties of the contractor play a major role in delaying construction projects, while Prasad, Vasugi (23) have also received the same result in the research they conducted in India…..”

Reviewer’s Comment 11: Please review this section to showcase and improve on the efforts put in previous sections. Especially regarding key factors and relationships. Avoid unnecessary ingredients in the conclusion section.

Authors’ Response to Comment 11: Comment has been well noted. This has been included in the revised manuscript from line 561 to 584.

“…..These factors were categorised into five groups namely, client/owner related factors, contractor related factors, consultant related factors, resource related factors and external factors. A questionnaire was developed incorporating questions relevant to these factors, which was effective in collecting data from the selected 163 Civil Engineers in Sri Lanka. The collected data were analysed using the RII. The factors were ranked accordingly to achieve the main objective of the study. The top ten project delaying factors were identified as, “Shortage of skilled subcontractors/suppliers”, “Shortage of labourers (Skilled, semi-skilled, unskilled)”, “Financial difficulties of contractors”, “Delay of delivering materials to site”, “Covid-19 pandemic situation”, “Fluctuation of material prices in the market”, “Poor planning and scheduling”, “Inadequate numbers of equipment”, “Poor site management, monitoring, and control” and “Delay in obtaining permissions/approvals from government”. According to the clients/owners and consultants, “Financial difficulties of contractors” was the most influential factor which delays a construction project. The contractors claimed that the “Shortage of skilled subcontractors/suppliers” was the most significant project delaying factor. Also, the most important group of factors was identified as the contractor related factors group. This was further confirmed by the qualitative database obtained by the respondents’ opinions. The most influential factor under the client/owner related factors was identified as the “Financial difficulties of clients”. “Shortage of skilled subcontractors/suppliers” ranked at the top of contractor related factors. “Delay in material and payment approval” and “Shortage of labours (Skilled, semi-skilled, unskilled)” were the most significant factors under consultant related factors and resource related factors respectively. When it comes to the external factors, “Covid-19 pandemic situation” was identified as the top ranked project delaying factor…..”

Reviewer’s Comment 12: Does this study have limitations? What recommendation do you have for future studies?

Authors’ Response to Comment 12: Comment has been well noted. Limitations of the study has been included in the revised manuscript from line 620 to 626. Recommendations has been included from line 628 to 643.

“Even though the study contributes largely to the construction industry in Sri Lanka, there could be some limitations which can be addressed in future research. The sample size could be expanded to be representative, compared to the large population of Civil Engineers employed in different types of construction work in Sri Lanka. This approach can assist for wide coverage and a comprehensive study to gain useful findings to avoid construction delays and related losses.”

“Based on the study's results, the financial difficulties of both contractors and clients should be evaluated for the successful completion of construction projects. Also, experienced subcontractors, suppliers and labourers should be available to avoid project delays. A skilled workforce is a must but lacking in the Sri Lankan setting. A transparent approach and unbiased selection of suitable contractors are mandatory to minimise root causes.

The timely completion of any construction project has a significant effect on the success of the project. Currently, the Sri Lankan construction industry is undergoing a booming phase, meaning meeting high demand and avoiding interruptions are vital. Therefore, the construction projects must be delivered on time without any delays, as it can also affect the country's overall economy. Therefore, the results of this study will be useful for the stakeholders to identify areas that deserve much focus and attention. Based on the findings of this research, the stakeholders can plan adequately, schedule, control and monitor the construction activities. Further, they can avoid losing revenue, undue delays, incurring additional costs and ensure credibility. The Sri Lankan government can use the findings of this study to foresee the reasons for delays in various construction projects and devise necessary mitigation measures.”

Reviewer’s Comment 13: Does this study have implications?

Authors’ Response to Comment 13: Comment has been well noted. Policy implications have been added in the revised manuscript from line 629 to 643. 

“Based on the study's results, the financial difficulties of both contractors and clients should be evaluated for the successful completion of construction projects. Also, experienced subcontractors, suppliers and labourers should be available to avoid project delays. A skilled workforce is a must but lacking in the Sri Lankan setting. A transparent approach and unbiased selection of suitable contractors are mandatory to minimise root causes.

The timely completion of any construction project has a significant effect on the success of the project. Currently, the Sri Lankan construction industry is undergoing a booming phase, meaning meeting high demand and avoiding interruptions are vital. Therefore, the construction projects must be delivered on time without any delays, as it can also affect the country's overall economy. Therefore, the results of this study will be useful for the stakeholders to identify areas that deserve much focus and attention. Based on the findings of this research, the stakeholders can plan adequately, schedule, control and monitor the construction activities. Further, they can avoid losing revenue, undue delays, incurring additional costs and ensure credibility. The Sri Lankan government can use the findings of this study to foresee the reasons for delays in various construction projects and devise necessary mitigation measures.”

Reviewer’s Comment 14: Revisit the reference list and make sure it conforms to the journals’ guide/format

Authors’ Response to Comment 14: Comment has been well noted. Reference list has been created as per the PLOS ONE journal guidelines. Please check the following link for more details. https://journals.plos.org/plosone/s/submission-guidelines/#loc-reference-style

---

## [Editor Report · Decision Letter 3]

15 Nov 2022

Factors influencing the timely completion of construction projects in Sri Lanka

PONE-D-22-23136R3

Dear Dr. Ruwan Jayathilaka,

We’re pleased to inform you that your manuscript has been judged scientifically suitable for publication and will be formally accepted for publication once it meets all outstanding technical requirements.

Kind regards,

Dr. Anu Sayal, Ph.D.

Academic Editor

PLOS ONE

---

## [Editor Report · Acceptance letter]

16 Nov 2022

PONE-D-22-23136R3 

Factors influencing the timely completion of construction projects in Sri Lanka 

Dear Dr. Jayathilaka:

I'm pleased to inform you that your manuscript has been deemed suitable for publication in PLOS ONE. Congratulations! Your manuscript is now with our production department. 

Kind regards, 

on behalf of

Dr. Anu Sayal 

Academic Editor

PLOS ONE